# Dynamic Analysis of Functionally Graded Porous (FGP) Elliptic Cylindrical Shell Based on Jacobi Polynomials Ritz Method

J. Lu [1], Q. Yang [1], Z. Meng [2], K. Yang [1], W. Xu [1] and C. Chiu [1,*]

[1] College of Mechanical & Electrical Engineering, Zaozhuang University, Zaozhuang 277160, China
[2] School of Mechanical Engineering, Shandong University, Jinan 250100, China
* Correspondence: cvinchiu@163.com

**Abstract:** The lightweight of structure is widely applied in industrial applications, and the conflict between both dynamic stability and structural lightweight is still prominent. In this paper, functionally graded porous (FGP) elliptic cylindrical shells and panels with general boundary conditions are analyzed to explore the effect of the FGP on dynamic performance. First, the FGP elliptic cylindrical shell and panel models are established. Therein, three kinds of porosity distribution are considered, including nonsymmetric, symmetric, and uniform distributions. The energy expressions of the FGP elliptic cylindrical shell and panel are established by the first-order shear deformation theory (FSDT). To simulate various boundary conditions, the artificial spring boundary technique is employed in this study. Then, the Jacobi orthogonal polynomials and Fourier series are adopted to express the admissible displacements. Finally, the accuracy of this model is verified by comparing it with open literature and ABAQUS software. Results show that the variations of the boundary conditions, linear springs, thickness ratio, and porosity have close relation with the dynamic performance of the structure by affecting the stiffness of the structure.

**Keywords:** functionally graded porous; dynamic analysis; elliptic cylindrical shell; elliptic cylindrical panel

## 1. Introduction

The shell and panel as structure elements have been widely employed in a variety of engineering fields due to their structural advantage [1]. The functionally graded porous (FGP) structures are novel structures with a graded distribution of internal pores in structures [2]. Furthermore, cylindrical shells and panels are usually exposed to complex environmental conditions. Functionally graded porous (FGM) materials, as a novel type of lightweight material, have both the performance of FGM materials and lightweight materials. Therefore, for better performance, shells and panels made of functionally grade materials have been widely studied.

Furthermore, the FGP material gives another way for the optimal design of structures on dynamic behavior [3]. In recent years, a significant number of articles have focused on the free vibrations, buckling, and bending behaviors of porous functionally graded structures [4–6]. By using the Timoshenko beam theory, Chen et al. [7] analyzed the elastic buckling and static bending characteristics of shear deformable FGP beams. Moreover, Ebrahimi and Zia et al. [8] presented the large-amplitude nonlinear vibration characteristics of the FGP Timoshenko core. Chen et al. [9] studied the nonlinear free vibration behavior of a shear deformable sandwich porous beam. Barati [10] studied the free vibrational behavior of FGP nanoshells using nonlocal strain gradient theory. It is shown that the vibrational behavior of the nanoshell is influenced by the porosity volume fraction, porosity distribution, nonlocal coefficient, strain gradient coefficient, boundary conditions, and radius-to-thickness ratio. Then, Wang and Wu [11] analyzed the dynamic behavior of an FGP cylindrical shell subject to different sets of immovable boundary conditions under free

vibration. A sinusoidal shear deformation theory, in conjunction with the Rayleigh–Ritz method, is employed to derive the governing equations associated with the free vibration of the circular cylindrical shell. Two types of graded porosity distributions in the thickness direction are considered. The study investigates the effects of FGP, boundary conditions, and geometrical parameters on free vibration characteristics of the FGP cylindrical shell. Mirjavadi et al. [12] analyzed the buckling and nonlinear vibration of the FGP nanobeam. The generalized differential quadrature method is used in conjunction with the iterative method to solve the equations. The parametric study is conducted to examine the effects of nonlinearity, porosity, sized effect on the vibration and buckling of porous nanobeam. In addition, Kim et al. [13] investigated bending, free vibration, and buckling response of FGP micro-plates using the classical and first order shear deformation plate theories. Numerical results of bending, free vibration, and buckling are presented to determine the effects of constituent material variation, microstructure-dependent size effects, and porosity distributions on the mechanical response of FGP micro-plates.

Different from the free vibration, bending, and buckling characteristics of the FGP structure, few investigations have been conducted on the dynamic characteristics of the FGP structure under forced vibration. Akbaş [14] investigated the forced vibration of a functionally graded deep beam with a porosity effect under a harmonic external distributed load. Two types of porosity distributions, even and uneven, were considered. A nonlinear three-parameter foundation model was employed to estimate the plate-foundation interactions. Huang et al. [15] studied the nonlinear free and forced vibrations of porous sigmoid functionally graded material plates resting on nonlinear elastic foundations. Mechanical properties of the functionally graded deep beam change in the thickness direction with porosity. Numerical results show that porosity plays a very important role in the dynamic responses of the functionally graded deep beam. As mentioned above, porosity plays a very important role in the dynamic characteristics of FGP structure under forced vibration.

To let the reader have a clear understanding of dynamic analysis, some investigations on FGMs are listed here. Bahri et al. [16] used the graded finite element method based on the Rayleigh–Ritz energy formulation to investigate the elastic behavior of laminated plates. Zhao and Choe et al. [17] studied the elastodynamic problems of functionally graded elliptic shells and panels with elastic constraints based on first-order shear deformation theory. Javanbakht et al. [18] presented a three-dimensional elasticity solution of a functionally graded piezoelectric shell panel under dynamic load.

In combination with these points above, the main purpose of this paper is to propose a modeling method for the dynamic analysis of the FGP elliptic cylindrical shells under free and forced vibrations. The structure of this paper is listed below: First, the models for FGP elliptic cylindrical shell and panel are established. Then, the energy expressions of the FGP cylindrical shell and panel are expressed, based on the artificial spring boundary technique, multi-segment technique, and modified variational method. A combination of Jacobi orthogonal polynomials and Fourier series is used to express the admissible displacements uniformly. With the constructed models, the accuracy of this model is verified by comparing it with literature and ABAQUS software. Then, the free vibration characteristic of the FGP structure is analyzed, and the forced response of the FGP elliptic cylindrical shell and panel is studied. At last, the numerical results obtained by the proposed method are discussed in Section 3.

## 2. Theoretical Formulations

### 2.1. FGP Elliptic Cylindrical Shell and Panel Model

To describe the structure of the FGP elliptic cylindrical shell and panel, a schematic diagram of the FGP elliptic cylindrical shell and panel is presented in Figure 1, where the FGP elliptic cylindrical shell is a closed cylinder, as shown in Figure 1a, and the FGP elliptic cylinder panel is an open cylinder, as shown in Figure 1b. A cylindrical coordinate system ($x$, $r$, and $\theta$) is employed to describe the geometrical characteristics, where $x$ denotes the axial direction, $r$ denotes the radial direction, and $\theta$ denotes the circumferential direction.



$L$ is the length of the FGP elliptic cylindrical shell and panel in the axial direction, and $a$ and $b$ denote the major semi-axis and minor semi-axis on the middle surface of the structure. The mean radius of the FGP elliptic cylindrical shell and panel $R_\theta(\theta)$ are expressed as [15]:

$$R_\theta(\theta) = \frac{a^2 b^2}{\sqrt{(a^2 \sin^2 \theta + b^2 \cos^2 \theta)^3}} \tag{1}$$

When $\theta < 360°$, FGP elliptic cylindrical panels can be described by Figure 1b; when $\theta = 360°$, FGP elliptic cylindrical shell can be described by Figure 1a. The purpose of this paper is to construct a unified modeling method for FGP cylindrical shell and panel; thus, different kinds of pore distributions need to be considered. Therefore, from Figure 1, three types of pore distribution are investigated in this paper. Type 1 is symmetric distribution; Type 2 is Nonsymmetric distribution; and Type 3 is Uniform distribution. For Type 1 and Type 2, the pore distribution varies in thickness direction. For Type 3, its pore distributes uniformly in the thickness direction. For a better description, the maximum value of Young's modulus is expressed as $E_1$, and the minimum value of Young's modulus is expressed as $E_2$. Expect that Type 3 has a uniform Young's modulus in the thickness direction, and Young's modulus for Type 1 and Type 2 both have a continuous nonlinear distribution in the thickness direction. For Type 1, the maximum value of Young's modulus achieves on the upper and the lower surfaces, and the minimum value of Young's modulus achieves on the middle surface. For Type 2, the maximum value of Young's modulus achieves on the upper surface, and the minimum value of Young's modulus achieves on the lower surface. The continuous nonlinear distribution of Young's modulus, shear modulus, and mass distribution in the thickness direction can be expressed as follows [6]:

$$\text{Type 1} \begin{cases} E(z) = E_1\left[1 - e_0 \cos\left(\frac{\pi z}{h}\right)\right] \\ G(z) = G_1\left[1 - e_0 \cos\left(\frac{\pi z}{h}\right)\right] \\ \rho(z) = \rho_1\left[1 - e_m \cos\left(\frac{\pi z}{h}\right)\right] \end{cases} \tag{2}$$

$$\text{Type 2} \begin{cases} E(z) = E_1\left[1 - e_0 \cos\left(\frac{\pi z}{2h} + \frac{\pi}{4}\right)\right] \\ G(z) = G_1\left[1 - e_0 \cos\left(\frac{\pi z}{2h} + \frac{\pi}{4}\right)\right] \\ \rho(z) = \rho_1\left[1 - e_m \cos\left(\frac{\pi z}{2h} + \frac{\pi}{4}\right)\right] \end{cases} \tag{3}$$

$$\text{Type 3} \begin{cases} E(z) = E_1(1 - e_0 \alpha) \\ G(z) = G_1(1 - e_0 \alpha) \\ \rho(z) = \rho_1\sqrt{1 - e_m \alpha} \end{cases} \tag{4}$$

The porous coefficient $e_0$ and density porous coefficient $e_m$ can be expressed as:

$$e_0 = 1 - \frac{E_2}{E_1} = 1 - \frac{G_2}{G_1}, 0 \le e_0 \le 1 \tag{5}$$

$$e_m = 1 - \frac{\rho_2}{\rho_1}, 0 \le e_m \le 1 \tag{6}$$

where $(E_1, G_1, \rho_1)$ are the maximum values of Young's modulus, shear modulus, and mass density, and $(E_2, G_2, \rho_2)$ represent the minimum values of Young's modulus, shear modulus, and mass density, respectively.

When $e_0 = e_m = 0$, it represents there have no holes in the materials, which means the case of solid materials; When $e_0 = e_m = 1$, it represents completely hollow, which is impossible in reality. According to the unique mechanical properties of porous foamed metal, the relationship between $e_0$, $e_m$, and coefficient $\alpha$ are defined as [7]:

$$e_m = 1 - \sqrt{1 - e_0} \tag{7}$$

$$\alpha = \frac{1}{e_0} - \frac{1}{e_0}\left(\frac{2}{\pi}\sqrt{1 - e_0} - \frac{2}{\pi} + 1\right)^2 \tag{8}$$

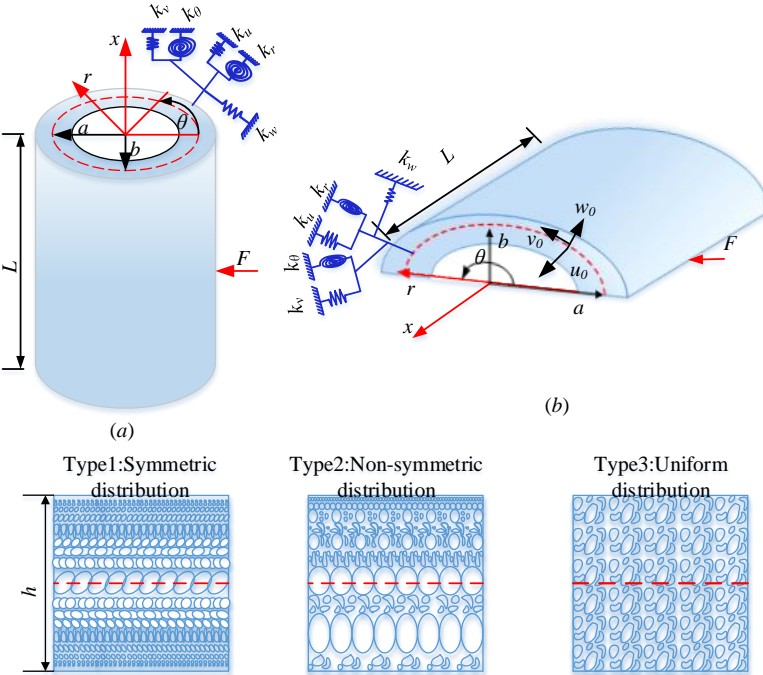

**Figure 1.** Schematic diagram of FGP elliptic cylindrical shell and panel. (**a**) FGP elliptic cylindrical shell, (**b**) elliptic cylindrical panel.

## 2.2. Energy Expression

For any point on the shell and panel, its displacement components are established based on first-order shear deformation shell theory (FSTD) as [17]:

$$u(x,\theta,z,t) = u_0(x,\theta,t) + z\phi_x(x,\theta,t)$$
$$v(x,\theta,z,t) = v_0(x,\theta,t) + z\phi_\theta(x,\theta,t)$$
$$w(x,\theta,z,t) = w_0(x,\theta,t)$$

(9)

where $u$, $v$, and $w$ represent the middle surface displacements of the FGP elliptic cylindrical structure element in the $x$, $\theta$, and $z$ directions, respectively. $\phi_x$, $\phi_\theta$ stand for the rotations of the normal to the middle surface in the $x$ and $\theta$ directions; $t$ denotes the time. The relationships between strains and displacements on the middle surface ($z = 0$) can be given as follows [17]:

$$\varepsilon_x^0 = \frac{\partial u_0}{\partial x}, \varepsilon_\theta^0 = \frac{1}{R_\theta}\left(\frac{\partial v_0}{\partial \theta} + w_0\right)$$

(10)

$$\chi_x = \frac{\partial \phi_x}{\partial x}, \chi_\theta = \frac{1}{R_\theta}\frac{\partial \phi_\theta}{\partial \theta}, \chi_{x\theta} = \frac{\partial \phi_\theta}{\partial x} + \frac{1}{R_\theta}\frac{\partial \phi_x}{\partial \theta}$$

(11)

$$\gamma_{x\theta}^0 = \frac{\partial v_0}{\partial x} + \frac{1}{R_\theta}\frac{\partial u_0}{\partial \theta}, \gamma_{xz}^0 = \frac{\partial w_0}{\partial x} + \phi_x, \gamma_{\theta z}^0 = \frac{1}{R_\theta}\left(\frac{\partial w_0}{\partial \theta} - v_0\right) + \phi_\theta$$

(12)

The force and moment resultants of the middle surface can be expressed as [18]:

$$\begin{bmatrix} N_x \\ N_\theta \\ N_{x\theta} \\ M_x \\ M_\theta \\ M_{x\theta} \end{bmatrix} = \begin{bmatrix} A_{11} & A_{12} & 0 & B_{11} & B_{12} & 0 \\ A_{12} & A_{11} & 0 & B_{12} & B_{11} & 0 \\ 0 & 0 & A_{66} & 0 & 0 & B_{66} \\ B_{11} & B_{12} & 0 & D_{11} & D_{12} & 0 \\ B_{12} & B_{11} & 0 & D_{12} & D_{11} & 0 \\ 0 & 0 & B_{66} & 0 & 0 & D_{66} \end{bmatrix} \begin{bmatrix} \varepsilon_x^0 \\ \varepsilon_\theta^0 \\ \varepsilon_{x\theta}^0 \\ \chi_x \\ \chi_\theta \\ \chi_{x\theta} \end{bmatrix}$$

(13)

$$\begin{bmatrix} Q_x \\ Q_\theta \end{bmatrix} = \begin{bmatrix} \kappa A_{66} & 0 \\ 0 & \kappa A_{66} \end{bmatrix} \begin{bmatrix} \gamma_{xz}^0 \\ \gamma_{\theta z}^0 \end{bmatrix}$$

(14)

where $N_x$, $N_\theta$, and $N_{x\theta}$ stand for the in-plane force resultants, $M_x$, $M_\theta$, and $M_{x\theta}$ denote moment resultants, and $Q_x$ and $Q_\theta$ express transverse shear force resultants. $\kappa$ stands for the shear correction factor. $A_{ij}$, $B_{ij}$, and $D_{ij}$ express the extensional, extensional-bending coupling, and bending stiffness, respectively. The above stiffness parameters can be defined as:

$$\left(A_{ij}, B_{ij}, D_{ij}\right) = \int_{-h/2}^{h/2} Q_{ij}(z)(1, z, z^2) dz \tag{15}$$

$$Q_{11}(z) = \frac{E(z)}{1 - \mu^2}, Q_{12}(z) = \frac{\mu E(z)}{1 - \mu^2}, Q_{66}(z) = \frac{E(z)}{2(1 + \mu)} \tag{16}$$

where the elastic constants $Q_{ij}$ *(z)* are functions of thickness coordinate *z*; $\mu$ is Poisson's ratio.

The strain energy (*U*) and kinetic energy (*T*) in FGP elliptic cylindrical shell or panel can be expressed as follows:

$$U = \frac{1}{2} \iint_s \left\{ \begin{array}{l} N_x \varepsilon_x^0 + N_\theta \varepsilon_\theta^0 + N_{x\theta} \varepsilon_{x\theta}^0 + M_x \chi_x \\ + M_\theta \chi_\theta + M_{x\theta} \chi_{x\theta} + Q_x \gamma_{xz}^0 + Q_\theta \gamma_{\theta z}^0 \end{array} \right\} dS \tag{17}$$

$$T = \frac{1}{2} \iint_s \left\{ \begin{array}{l} I_0 \left[ \dot{U}^2 + \dot{V}^2 + \dot{W}^2 \right] + 2I_1 \left( \dot{U} \dot{\phi}_x + \dot{V} \dot{\phi}_\theta \right) \\ + I_2 \left[ \dot{\phi}_x^2 + \dot{\phi}_\theta^2 \right] \end{array} \right\} dS \tag{18}$$

where the inertia terms can be expressed with the following equations:

$$(I_0, I_1, I_2) = \int_{-h/2}^{h/2} \rho(z)(1, z^1, z^2) dz \tag{19}$$

For the structural strain energy, it consists of three parts, including stretching ($U_s$), bending ($U_b$), and bending stretching ($U_{bs}$) coupling energy expressions, which can be expressed as [18]

$$U_s = \frac{1}{2} \iint_S \left\{ \begin{array}{l} A_{11} \left( \frac{1}{A} \frac{\partial u_0}{\partial x} + \frac{v_0}{AB} \frac{\partial A}{\partial \theta} + \frac{w_0}{R_x} \right)^2 + \kappa A_{66} \left( \phi_x + \frac{1}{A} \frac{\partial w_0}{\partial x} \right)^2 \\ + A_{11} \left( \frac{1}{B} \frac{\partial v_0}{\partial \theta} + \frac{u_0}{AB} \frac{\partial B}{\partial x} + \frac{w_0}{R_\theta} \right)^2 + \kappa A_{66} \left( \phi_\theta + \frac{1}{B} \frac{\partial w_0}{\partial \theta} \right)^2 \\ + 2A_{12} \left( \frac{1}{A} \frac{\partial u_0}{\partial x} + \frac{v_0}{AB} \frac{\partial A}{\partial \theta} + \frac{w_0}{R_x} \right) \left( \frac{1}{B} \frac{\partial v_0}{\partial \theta} + \frac{u_0}{AB} \frac{\partial B}{\partial x} + \frac{w_0}{R_\theta} \right) \\ + A_{66} \left( \frac{1}{A} \frac{\partial v_0}{\partial x} - \frac{u_0}{AB} \frac{\partial A}{\partial \theta} + \frac{1}{B} \frac{\partial u_0}{\partial \theta} - \frac{v_0}{AB} \frac{\partial B}{\partial x} \right)^2 \end{array} \right\} dS \tag{20}$$

$$U_{bs} = \iint_S \left\{ \begin{array}{l} B_{11} \left[ \begin{array}{l} \left( \frac{1}{A} \frac{\partial u_0}{\partial x} + \frac{v_0}{AB} \frac{\partial A}{\partial \theta} + \frac{w_0}{R_x} \right) \left( \frac{1}{A} \frac{\partial \phi_x}{\partial x} + \frac{\phi_\theta}{AB} \frac{\partial A}{\partial \theta} \right) + \\ \left( \frac{1}{B} \frac{\partial v_0}{\partial \theta} + \frac{u_0}{AB} \frac{\partial B}{\partial x} + \frac{w_0}{R_\theta} \right) \left( \frac{1}{B} \frac{\partial \phi_\theta}{\partial \theta} + \frac{\phi_x}{AB} \frac{\partial B}{\partial x} \right) \end{array} \right] \\ + B_{12} \left[ \begin{array}{l} \left( \frac{1}{A} \frac{\partial u_0}{\partial x} + \frac{v_0}{AB} \frac{\partial A}{\partial \theta} + \frac{w_0}{R_x} \right) \left( \frac{1}{B} \frac{\partial \phi_\theta}{\partial \theta} + \frac{\phi_x}{AB} \frac{\partial B}{\partial x} \right) + \\ \left( \frac{1}{B} \frac{\partial v_0}{\partial \theta} + \frac{u_0}{AB} \frac{\partial B}{\partial x} + \frac{w_0}{R_\theta} \right) \left( \frac{1}{A} \frac{\partial \phi_x}{\partial x} + \frac{\phi_\theta}{AB} \frac{\partial A}{\partial \theta} \right) \end{array} \right] \\ + B_{66} \left[ \begin{array}{l} \left( \frac{1}{A} \frac{\partial v_0}{\partial x} - \frac{u_0}{AB} \frac{\partial A}{\partial \theta} + \frac{1}{B} \frac{\partial u_0}{\partial \theta} - \frac{v_0}{AB} \frac{\partial B}{\partial \alpha} \right) \\ \left( \frac{1}{A} \frac{\partial \phi_\theta}{\partial x} - \frac{\phi_x}{AB} \frac{\partial A}{\partial \theta} + \frac{1}{B} \frac{\partial \phi_x}{\partial \theta} - \frac{\phi_\beta}{AB} \frac{\partial B}{\partial x} \right) \end{array} \right] \end{array} \right\} dS \tag{21}$$

$$U_b = \frac{1}{2} \iint_S \left\{ \begin{array}{l} D_{11} \left( \frac{1}{A} \frac{\partial \phi_x}{\partial x} + \frac{\phi_\theta}{AB} \frac{\partial A}{\partial \theta} \right)^2 + D_{11} \left( \frac{1}{B} \frac{\partial \phi_\theta}{\partial \theta} + \frac{\phi_x}{AB} \frac{\partial B}{\partial x} \right)^2 \\ + 2D_{12} \left( \frac{1}{A} \frac{\partial \phi_x}{\partial x} + \frac{\phi_\theta}{AB} \frac{\partial A}{\partial \theta} \right) \left( \frac{1}{B} \frac{\partial \phi_\theta}{\partial \theta} + \frac{\phi_x}{AB} \frac{\partial B}{\partial x} \right) \\ + D_{66} \left( \frac{1}{A} \frac{\partial \phi_\theta}{\partial x} - \frac{\phi_x}{AB} \frac{\partial A}{\partial \theta} + \frac{1}{B} \frac{\partial \phi_x}{\partial \theta} - \frac{\phi_\theta}{AB} \frac{\partial B}{\partial x} \right)^2 \end{array} \right\} dS \tag{22}$$

where the quantities *A* and *B* are the Lamé parameters.

Afterward, the potential energies can be obtained by using the artificial spring boundary technique:

$$
\begin{aligned}
U_{sp} = &\tfrac{1}{2}\int_{-h/2}^{h/2}\int_0^\theta \left\{ \Sigma k_{x_0}^i (u_i)^2 + \Sigma k_{x_1}^i (u_i)^2 \right\} d\theta dz \\
&+ \tfrac{1}{2}\int_{-h/2}^{h/2}\int_0^L \left\{ \Sigma k_{\theta_0}^i (u_i)^2 + \Sigma k_{\theta_1}^i (u_i)^2 \right\} R dx dz
\end{aligned}
\tag{23}
$$

where $k^i u_j$ ($i = u$, $v$, $w$, $\phi_x$, $\phi_\theta$) denotes the equations describing general elastic supported FGP structure elements; $x_0$, $x_1$, $\theta_0$, $\theta_1$ represent the boundary of the structure, which $x_0 = 0$, $x_1 = L$, $\theta_0 = 0$, $\theta_1 = \theta$. Need to be mentioned that, in this equation, only FGP elliptic cylindrical panel is considered. For FGP elliptic cylindrical shell, which can be described as $\theta = 2\pi$. The kinematic and physical compatibility conditions between $\theta = 0$ and $\theta = 2\pi$ should be implemented. For this, we adopt the coupling spring technique, and the potential energies stored in these springs can be written as:

$$
\begin{aligned}
U_{cp}^{ij} = &\tfrac{1}{2}\int_{-h/2}^{h/2}\int_{L_i}^{L_{i+1}} \left\{ \begin{array}{l} k_{uc}^u \left(u_{ij} - u_{ij+1}\right)^2 + k_{vc}^v \left(v_{ij} - v_{ij+1}\right)^2 + \\ k_{wc}^w \left(w_{ij} - w_{ij+1}\right)^2 + K_{xc}^x \left(\phi_{x_{ij}} - \phi_{x_{ij+1}}\right)^2 \\ + K_{\theta c}^\theta \left(\phi_{\theta_{ij}} - \phi_{\theta_{ij+1}}\right)^2 \end{array} \right\} dx dz \\
&+ \tfrac{1}{2}\int_{-h/2}^{h/2}\int_{\theta_j}^{\theta_{j+1}} \left\{ \begin{array}{l} k_{uc}^u \left(u_{ij} - u_{i+1j}\right)^2 + k_{vc}^v \left(v_{ij} - v_{i+1j}\right)^2 + \\ k_{wc}^w \left(w_{ij} - w_{i+1j}\right)^2 + K_{xc}^x \left(\phi_{x_{ij}} - \phi_{x_{i+1j}}\right)^2 \\ + K_{\theta c}^\theta \left(\phi_{\theta_{ij}} - \phi_{\theta_{i+1j}}\right)^2 \end{array} \right\} R_\theta d\theta dz
\end{aligned}
\tag{24}
$$

### 2.3. Admissible Displacements and Solving Process

In this section, a combination of Jacobi orthogonal polynomials and Fourier series are employed in this study. The displacement and rotation components of shell segments are uniformly expressed as follows:

$$
\begin{aligned}
u_0 &= \sum_{m=0}^M \sum_{n=0}^N U_{mn} P_m^{(\alpha,\beta)}(x) P_n^{(\alpha,\beta)}(\theta) e^{i\omega t}; \\
v_0 &= \sum_{m=0}^M \sum_{n=0}^N V_{mn} P_m^{(\alpha,\beta)}(x) P_n^{(\alpha,\beta)}(\theta) e^{i\omega t}; \\
w_0 &= \sum_{m=0}^M \sum_{n=0}^N W_{mn} P_m^{(\alpha,\beta)}(x) P_n^{(\alpha,\beta)}(\theta) e^{i\omega t}; \\
\phi_x &= \sum_{m=0}^M \sum_{n=0}^N \Phi_{xmn} P_m^{(\alpha,\beta)}(x) P_n^{(\alpha,\beta)}(\theta) e^{i\omega t}; \\
\phi_\theta &= \sum_{m=0}^M \sum_{n=0}^N \Phi_{\theta mn} P_m^{(\alpha,\beta)}(x) P_n^{(\alpha,\beta)}(\theta) e^{i\omega t};
\end{aligned}
\tag{25}
$$

where $U_{mn}$, $V_{mn}$, $W_{mn}$, $\Phi_{xmn}$, and $\Phi_{\theta mn}$ denote the corresponding Jacobi expansion coefficients; $P$ is the Jacobi polynomial of order $m$ of the displacement component on the middle surface of elliptic cylindrical shell and panel; $\omega$ stands for the angular frequency; $t$ is the time variable; $M$ denotes the highest order in admissible function. By choosing different combinations of $\alpha$ and $\beta$, the special form can be obtained: (a) $\alpha = \beta = -1/2$, Chepyshev polynomials of the first type; (b) $\alpha = \beta = 1/2$, Chebyshev polynomials of the second type; (c) $\alpha = \beta = 0$, Legendre polynomials; (d) $\alpha = \beta$, Gegenbauer polynomials. Therefore, by adopting the Jacobi polynomials, the admissible displacement functions may be more generalized.

The Lagrange equation $L$ of the FGP structure can be expressed as:

$$L = T - U - U_{cp} - U_{sp} \tag{26}$$

Then, the derivation of Lagrange equations can be obtained:

$$\frac{\partial L}{\partial U_m} = \frac{\partial L}{\partial V_m} = \frac{\partial L}{\partial W_m} = \frac{\partial L}{\partial \Phi_{x_m}} = \frac{\partial L}{\partial \Phi_{\theta_m}} = 0 \tag{27}$$

The equations can be organized into the matrix form as follows:

$$\left( K - \omega^2 M \right) E = 0 \tag{28}$$

where K stands for the stiffness matric for the FGP structure, and M expresses the mass matrix for the FGP structure; E means the unknown vector, and can be expressed as $E = [U_m, \ V_m, \ W_m, \Phi_{x_m}, \ \Phi_{\theta_m}]^T$.

Subsequently, considering the external force effect, the dynamic characteristic equation of the FGP structure can be obtained as follows:

$$(K - \omega^2 M) E = F \tag{29}$$

where F is the external excitation force.

In addition, the structural dynamic equilibrium equation of the FGP elliptic cylindrical shell is defined as

$$M\ddot{u} + C\dot{u} + Ku = F \tag{30}$$

where $\ddot{u}$ expresses the acceleration vector of the FGP elliptic cylindrical shell; $\dot{u}$ means the velocity vector of the FGP elliptic cylindrical shell; $u$ stands for the displacement vector of the FGP elliptic cylindrical shell; C expresses the damping matrix, $C = \gamma_1 M + \gamma_2 K$, where $\gamma_1$ and $\gamma_2$ mean the Rayleigh damping coefficients, $\gamma_1 = 2 \left( \frac{\zeta_2}{\omega_2^3} - \frac{\zeta_1}{\omega_1^3} \right) / \left( \frac{1}{\omega_2^2} - \frac{1}{\omega_1^2} \right)$ and $\gamma_2 = 4 \left( \frac{\zeta_2 - \zeta_1}{\omega_2^2 - \omega_1^2} \right)$, where $\zeta_1$, $\zeta_2$ are the damping ratios [19].

Based on the Newmark–Beta method [20], the transient response of the FGP elliptic cylindrical shell is expressed as

$$\dot{u}_{t+\Delta t} = \dot{u}_t + \left[ (1 - \delta)\ddot{u}_t + \delta \ddot{u}_{t+\Delta t} \right] \Delta t \tag{31}$$

$$u_{t+\Delta t} = u_t + \dot{u}_t \Delta t + \left[ \left( \frac{1}{2} - \alpha \right) \ddot{u}_t + \alpha \ddot{u}_{t+\Delta t} \right] \Delta t^2 \tag{32}$$

where $\alpha$ and $\delta$ can be set as 1/4 and 1/2 (unconditionally stable second-order method), respectively.

## 3. Numerical Results and Discussion

In this section, the correction of this method is verified first. Then, the free vibration characteristics of the FGP cylindrical shell and panel are studied. Next, the steady-state response characteristics in forced response and the transient response characteristics of forced response of the FGP cylindrical shell and panel are studied. For the numerical studies of the FGP elliptic cylindrical shell and panel below, it is important to mention that the calculation conditions are set as follows: The coupling springs of the FGP elliptic cylindrical shell and panel segments are set as 1014 N/m. The number of segments of the FGP elliptic cylindrical shell and panel segments is set to 6; Jacobi expansion items are set to 8. As for calculation boundary conditions, both the classical and elastic boundaries are considered. There are three types of classical boundary conditions and three types of elastic boundary conditions. Classical boundary conditions such as clamping, simple support, and free boundary conditions are considered here. In order to simplify the narration, clamping, simple support, and free boundary conditions are represented by letters C, S, and F, respectively. The elastic boundary is denoted as E1, E2, and E3, respectively. The boundary conditions are defined as follows (take the L = 0 m boundary as an example)

Clamped cases (C): $k_0^{u_0} = k_0^{v_0} = k_0^{w_0} = k_0^{\phi_x} = k_0^{\phi_y} = 10^{14}$, Free cases (F): $k_0^{u_0} = k_0^{v_0} = k_0^{w_0} = k_0^{\phi_x} = k_0^{\phi_y} = 0$, Simply support cases (S): $k_0^{u_0} = k_0^{\phi_x} = 0, k_0^{v_0} = k_0^{w_0} = k_0^{\phi_y} = 10^{14}$, Elastic constraint 1 (E1): $k_0^{u_0} = 10^8, k_0^{v_0} = k_0^{w_0} = k_0^{\phi_x} = k_0^{\phi_y} = 10^{14}$, Elastic constraint 2 (E2): $k_0^{u_0} = k_0^{\phi_x} = 10^8, k_0^{w_0} = k_0^{\phi_x} = k_0^{\phi_y} = 10^{14}$, Elastic constraint 3 (E3): $k_0^{u_0} = k_0^{v_0} = k_0^{\phi_x} = 10^8, k_0^{w_0} = k_0^{\phi_y} = 10^{14}$.

### 3.1. Dynamic Characteristics of FGP Elliptic Cylindrical Shell under Free Vibration

In this section, the free vibration characteristics of the FGP elliptic cylindrical shell and panel were studied. Before beginning the studies, the correctness of the model needs to be verified. To verify the correctness of this method, a comparative study was carried out.

Table 1 shows the comparative study of Type 1 FGP elliptic cylindrical shell and panel with various boundary conditions. The study of Type 1 FGP elliptic cylinder shells with various boundary conditions is compared with Guan et al. [21]. It needs to be mentioned that the comparison example only considered a special case of FGP elliptic cylinder shell and panel, in which the length of the long radius and short radius is equal ($a = b$). The preset parameters are as follows: For FGP elliptic cylindrical shell, the geometrical parameters are $L = 2$ m, $a = 1$ m, $b = 1$ m, and $h = 0.1$ m. For FGP elliptic cylindrical panel, the geometrical parameters are $L = 5$ m, $a = 1$ m, $b = 1$ m, $\theta = 2\pi/3$, and $h = 0.1$ m. The material parameters of both are $\rho_1 = 7850$ kg/m$^3$, $E_1 = 200$ GPa, and $\mu = 0.3$. The object of comparison is the dimensionless frequency parameter $\Omega = \omega b \sqrt{\rho_1/E_1}$. Compared with the first four frequencies $\Omega$, it is found that the dimensionless frequency parameter $\Omega$ in the simulation of elliptic shells in this paper is slightly smaller than the dimensionless frequency parameters obtained in the reference. For example, the fourth frequency of the FGP elliptic cylindrical panel with CCCC is slightly smaller in this paper; the second and fourth frequency of the FGP elliptic cylindrical shell with CF boundary condition is slightly smaller in this paper. The calculation results of the model in this paper are in good agreement with those in the reference, and the maximum error is less than 1%. So the correction of this method is verified.

After verifying the correctness of the model, it is possible to carry out new computational studies using this model. Table 2 shows the frequency parameters of the FGP elliptic cylindrical shell and panel under different classical and elastic boundary conditions. The preset parameters are given as follows: For FGP elliptic cylindrical shell, the geometrical parameters are $L = 5$ m, $a = 1$ m, $b = 2$ m, and $h = 0.1$ m. For FGP elliptic cylindrical panel, the geometrical parameters are $L = 5$ m, $a = 1$ m, $b = 2$ m, $\theta = \pi$, and $h = 0.1$ m. The material parameters of both are $\rho_1 = 2702$ kg/m$^3$, $E_1 = 70$ GPa, and $\mu = 0.3$. It is discovered in Table 2 that the frequency parameter with complete clamped boundary conditions is maximum. This kind of phenomenon is directly related to the specific stiffness value. To better describe this phenomenon, the influence of boundary parameters on structural characteristics is further studied. The specific study with the effect of boundary elastic parameters on FGP elliptic cylindrical shell and panel vibration characteristics is illustrated in Figure 2. The boundary conditions in Figure 2 are defined as follows: The boundary condition at $x = 0$ is a solid support boundary constraint. The boundary condition at $x = L$ is an elastic constraint, which only changes one type of spring each time and keeps the other four as $10^{14}$. It can be intuitively found in Figure 2 that the frequency parameter of this structure increases rapidly with the increase of boundary stiffness, and when the value of boundary stiffness exceeds a threshold, frequency parameter region convergence occurs. The boundary can be seen as solid support at this time. Hence, the vibration characteristics of the structure can be regulated by changing the value of boundary stiffness. In addition, we can also intuitively find that linear spring $k_u$ is the most sensitive to the influence of vibration characteristics of the structure. The influence of the rotation spring is very little on the vibration behaviors of the structure.

**Table 1.** Comparison of the first four frequencies $\Omega = \omega b \sqrt{\rho_1/E_1}$ of Type 1 FGP elliptic cylindrical shell with various boundary conditions.

| $e_0$ | Mode | FGP Elliptic Cylindrical Panel | | | | | | FGP Elliptic Cylindrical Shell | | | | | |
|---|---|---|---|---|---|---|---|---|---|---|---|---|---|
| | | CCCC | | SSSS | | CSCF | | CC | | SS | | CF | |
| | | Guan [21] | Present | Guan [21] | Present | Guan [21] | Present | Guan [21] | Present | Guan [21] | Present | Guan [21] | Present |
| 0.2 | 1 | 0.569 | 0.569 | 0.497 | 0.497 | 0.241 | 0.241 | 0.423 | 0.422 | 0.386 | 0.385 | 0.171 | 0.171 |
| | 2 | 0.695 | 0.695 | 0.890 | 0.590 | 0.262 | 0.262 | 0.440 | 0.440 | 0.406 | 0.406 | 0.247 | 0.246 |
| | 3 | 0.911 | 0.911 | 0.825 | 0.825 | 0.449 | 0.449 | 0.545 | 0.543 | 0.517 | 0.515 | 0.284 | 0.284 |
| | 4 | 0.938 | 0.937 | 0.844 | 0.844 | 0.517 | 0.517 | 0.613 | 0.613 | 0.594 | 0.594 | 0.432 | 0.430 |
| 0.4 | 1 | 0.547 | 0.547 | 0.480 | 0.480 | 0.232 | 0.232 | 0.406 | 0.406 | 0.368 | 0.367 | 0.165 | 0.165 |
| | 2 | 0.668 | 0.668 | 0.563 | 0.563 | 0.252 | 0.252 | 0.424 | 0.424 | 0.388 | 0.388 | 0.237 | 0.237 |
| | 3 | 0.877 | 0.877 | 0.797 | 0.797 | 0.433 | 0.432 | 0.523 | 0.521 | 0.494 | 0.492 | 0.274 | 0.274 |
| | 4 | 0.903 | 0.902 | 0.810 | 0.810 | 0.497 | 0.496 | 0.591 | 0.591 | 0.572 | 0.572 | 0.414 | 0.412 |

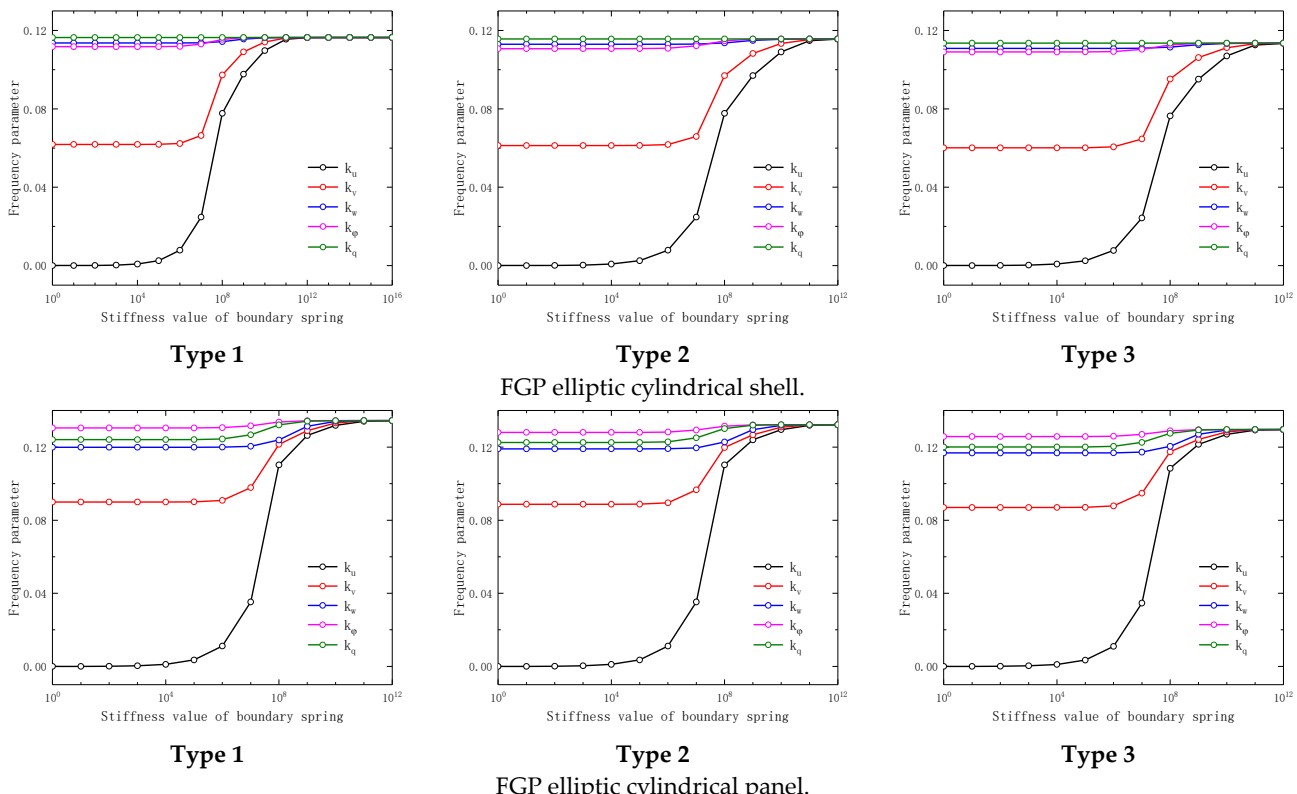

**Figure 2.** Effect of boundary elastic parameters on vibration characteristics of FGP elliptic cylindrical shell and panel.

**Table 2.** Frequencies $\Omega = \omega b \sqrt{\rho_1 / E_1}$ of FGP elliptic cylindrical shell and panel for different boundary conditions and porosity distribution.

| Type | $e_0$ | FGP Elliptic Cylindrical Shell | | | | | | FGP Elliptic Cylindrical Panel | | | | | |
| --- | --- | --- | --- | --- | --- | --- | --- | --- | --- | --- | --- | --- | --- |
| | | CC | CF | CS | E1E2 | E1E3 | E2E3 | CCCC | CFCF | CSCF | E1E2E1E2 | E1E3E1E3 | E2E3E2E3 |
| Type 1 | 0.1 | 0.118 | 0.046 | 0.116 | 0.104 | 0.076 | 0.090 | 0.135 | 0.073 | 0.078 | 0.112 | 0.098 | 0.119 |
| | 0.2 | 0.116 | 0.046 | 0.114 | 0.103 | 0.078 | 0.090 | 0.134 | 0.072 | 0.078 | 0.111 | 0.098 | 0.119 |
| | 0.3 | 0.115 | 0.046 | 0.112 | 0.101 | 0.078 | 0.089 | 0.134 | 0.072 | 0.077 | 0.111 | 0.097 | 0.118 |
| | 0.4 | 0.113 | 0.045 | 0.111 | 0.100 | 0.078 | 0.089 | 0.133 | 0.071 | 0.076 | 0.110 | 0.097 | 0.117 |
| Type 2 | 0.1 | 0.118 | 0.046 | 0.115 | 0.104 | 0.076 | 0.090 | 0.134 | 0.073 | 0.078 | 0.112 | 0.098 | 0.119 |
| | 0.2 | 0.116 | 0.045 | 0.113 | 0.102 | 0.078 | 0.089 | 0.132 | 0.071 | 0.077 | 0.110 | 0.097 | 0.117 |
| | 0.3 | 0.114 | 0.044 | 0.111 | 0.100 | 0.077 | 0.088 | 0.130 | 0.070 | 0.075 | 0.109 | 0.095 | 0.116 |
| | 0.4 | 0.112 | 0.044 | 0.109 | 0.098 | 0.076 | 0.087 | 0.127 | 0.069 | 0.074 | 0.107 | 0.094 | 0.114 |
| Type 3 | 0.1 | 0.117 | 0.046 | 0.114 | 0.103 | 0.076 | 0.089 | 0.133 | 0.072 | 0.077 | 0.111 | 0.097 | 0.118 |
| | 0.2 | 0.114 | 0.044 | 0.111 | 0.100 | 0.076 | 0.087 | 0.130 | 0.070 | 0.075 | 0.108 | 0.095 | 0.115 |
| | 0.3 | 0.110 | 0.043 | 0.108 | 0.097 | 0.075 | 0.086 | 0.126 | 0.068 | 0.073 | 0.105 | 0.093 | 0.112 |
| | 0.4 | 0.107 | 0.042 | 0.105 | 0.094 | 0.073 | 0.083 | 0.122 | 0.066 | 0.071 | 0.102 | 0.090 | 0.109 |

The frequency parameters of the FGP elliptic cylindrical shell and panel under different radius ratio is given in Table 3. The boundary condition of this study is set as CS. The porosity parameter varies from 0.2 to 0.7. The geometrical parameters and material parameters are given as follows: For FGP elliptic cylindrical shell, the geometrical parameters are $L = 5$ m, $b = 1$ m, and $h = 0.1$ m. For FGP elliptic cylindrical panel, the geometrical parameters are $L = 5$ m, $a = 1$ m, $\theta = \pi$, and $h = 0.1$ m. The material parameters of both are $\rho_1 = 2702$ kg/m$^3$, $E_1 = 70$ GPa, and $\mu = 0.3$. The radius ratio is defined as four groups, respectively, as follows: $a/b = 0.5, 1, 1.5, 2$. It can be seen from Table 3 that the frequency

parameter of the FGP elliptic cylindrical shell and panel decreases with the increase of radius ratio. This phenomenon is because the structure mass matrix coefficient increases with the growth of the radius ratio, and thus the frequency parameters are reduced.

**Table 3.** Frequencies $\Omega = \omega b \sqrt{\rho_1/E_1}$ of FGP elliptic cylindrical shell and panel for different radius ratios $a/b$ and porosity distribution with CS boundary condition.

| Type | a/b | FGP Elliptic Cylindrical Shell | | | | | | FGP Elliptic Cylindrical Panel | | | | | |
|---|---|---|---|---|---|---|---|---|---|---|---|---|---|
| | | $e_0 = 0.2$ | $e_0 = 0.3$ | $e_0 = 0.4$ | $e_0 = 0.5$ | $e_0 = 0.6$ | $e_0 = 0.7$ | $e_0 = 0.2$ | $e_0 = 0.3$ | $e_0 = 0.4$ | $e_0 = 0.5$ | $e_0 = 0.6$ | $e_0 = 0.7$ |
| Type 1 | 0.5 | 0.161 | 0.160 | 0.159 | 0.157 | 0.154 | 0.152 | 0.250 | 0.248 | 0.246 | 0.245 | 0.244 | 0.244 |
| | 1 | 0.159 | 0.157 | 0.155 | 0.154 | 0.152 | 0.151 | 0.249 | 0.247 | 0.244 | 0.241 | 0.239 | 0.237 |
| | 1.5 | 0.139 | 0.137 | 0.135 | 0.133 | 0.132 | 0.130 | 0.194 | 0.193 | 0.192 | 0.191 | 0.191 | 0.192 |
| | 2 | 0.114 | 0.112 | 0.111 | 0.109 | 0.108 | 0.107 | 0.134 | 0.134 | 0.133 | 0.132 | 0.132 | 0.132 |
| Type 2 | 0.5 | 0.157 | 0.154 | 0.151 | 0.147 | 0.143 | 0.137 | 0.246 | 0.242 | 0.237 | 0.232 | 0.225 | 0.219 |
| | 1 | 0.157 | 0.154 | 0.151 | 0.147 | 0.143 | 0.139 | 0.247 | 0.243 | 0.238 | 0.233 | 0.227 | 0.221 |
| | 1.5 | 0.138 | 0.135 | 0.132 | 0.129 | 0.126 | 0.123 | 0.190 | 0.187 | 0.183 | 0.178 | 0.173 | 0.167 |
| | 2 | 0.113 | 0.111 | 0.109 | 0.106 | 0.104 | 0.101 | 0.132 | 0.130 | 0.127 | 0.124 | 0.121 | 0.117 |
| Type 3 | 0.5 | 0.154 | 0.150 | 0.145 | 0.140 | 0.134 | 0.127 | 0.242 | 0.235 | 0.227 | 0.219 | 0.210 | 0.199 |
| | 1 | 0.155 | 0.150 | 0.145 | 0.140 | 0.134 | 0.128 | 0.242 | 0.235 | 0.228 | 0.219 | 0.210 | 0.200 |
| | 1.5 | 0.136 | 0.132 | 0.127 | 0.123 | 0.118 | 0.112 | 0.186 | 0.181 | 0.175 | 0.169 | 0.162 | 0.154 |
| | 2 | 0.111 | 0.108 | 0.105 | 0.101 | 0.097 | 0.092 | 0.130 | 0.126 | 0.122 | 0.117 | 0.112 | 0.107 |

Table 4 gives the calculation results of the frequency parameters of the FGP elliptic cylindrical shell and panel with different length ratios. The boundary condition is set as CS. The porosity parameter varies from 0.2 to 0.7. The geometrical parameters and material parameters of this study are the same as those in Table 2, except the length coefficient $L$ is different. The length ratio is defined as four groups, respectively, as follows: $L/a = 2$, 3, 4, 5. From Table 5, the frequency parameter of the FGP elliptic cylindrical shell and panel decreases with the increase in length ratio. The reason for this phenomenon is that the structure mass matrix coefficient increase with the increase of length, and thus the frequency parameters are reduced.

**Table 4.** Frequencies $\Omega = \omega b \sqrt{\rho_1/E_1}$ of FGP elliptic cylindrical shell and panel for different length ratios and porosity distribution.

| Type | L/a | FGP Elliptic Cylindrical Shell | | | | | | FGP Elliptic Cylindrical Panel | | | | | |
|---|---|---|---|---|---|---|---|---|---|---|---|---|---|
| | | $e_0 = 0.2$ | $e_0 = 0.3$ | $e_0 = 0.4$ | $e_0 = 0.5$ | $e_0 = 0.6$ | $e_0 = 0.7$ | $e_0 = 0.2$ | $e_0 = 0.3$ | $e_0 = 0.4$ | $e_0 = 0.5$ | $e_0 = 0.6$ | $e_0 = 0.7$ |
| Type 1 | 2 | 0.243 | 0.240 | 0.238 | 0.236 | 0.235 | 0.234 | 0.273 | 0.270 | 0.269 | 0.267 | 0.266 | 0.266 |
| | 3 | 0.173 | 0.171 | 0.169 | 0.168 | 0.167 | 0.166 | 0.189 | 0.187 | 0.185 | 0.184 | 0.183 | 0.183 |
| | 4 | 0.138 | 0.137 | 0.136 | 0.135 | 0.134 | 0.132 | 0.153 | 0.152 | 0.151 | 0.150 | 0.149 | 0.149 |
| | 5 | 0.114 | 0.112 | 0.111 | 0.109 | 0.108 | 0.107 | 0.134 | 0.134 | 0.133 | 0.132 | 0.132 | 0.132 |
| Type 2 | 2 | 0.239 | 0.235 | 0.230 | 0.225 | 0.220 | 0.213 | 0.269 | 0.264 | 0.259 | 0.253 | 0.247 | 0.239 |
| | 3 | 0.170 | 0.167 | 0.163 | 0.160 | 0.155 | 0.150 | 0.186 | 0.183 | 0.179 | 0.175 | 0.171 | 0.166 |
| | 4 | 0.136 | 0.133 | 0.131 | 0.127 | 0.124 | 0.120 | 0.151 | 0.148 | 0.145 | 0.142 | 0.139 | 0.134 |
| | 5 | 0.113 | 0.111 | 0.109 | 0.106 | 0.104 | 0.101 | 0.132 | 0.130 | 0.127 | 0.124 | 0.121 | 0.117 |
| Type 3 | 2 | 0.235 | 0.228 | 0.221 | 0.213 | 0.204 | 0.194 | 0.263 | 0.256 | 0.247 | 0.238 | 0.228 | 0.217 |
| | 3 | 0.167 | 0.162 | 0.157 | 0.151 | 0.145 | 0.138 | 0.182 | 0.177 | 0.171 | 0.165 | 0.158 | 0.151 |
| | 4 | 0.134 | 0.130 | 0.126 | 0.121 | 0.116 | 0.110 | 0.148 | 0.144 | 0.139 | 0.134 | 0.129 | 0.122 |
| | 5 | 0.111 | 0.108 | 0.105 | 0.101 | 0.097 | 0.092 | 0.130 | 0.126 | 0.122 | 0.117 | 0.112 | 0.107 |

Furthermore, as for the FGP elliptic cylindrical panel, the opening angle also influences vibration characteristics. The specific research result is shown in Table 5. The geometrical parameters are consistent with Table 4, except the opening angle is different. It can be seen from Table 4 that the frequency parameter decreases gradually with the increase of the opening angle. The reason for this phenomenon is that the structure mass matrix coefficient increase with the increase of the opening angle, and thus the frequency parameters are reduced. And this phenomenon is especially obvious when the opening angle is small. From Table 2 to Table 5, it can be seen that the frequency parameter of the FGP elliptic cylindrical shell and panel with Type 1 porosity distribution is the largest, and the frequency parameter of the FGP elliptic cylindrical shell and panel with Type 3 porosity distribution is the smallest. In addition, no matter which kind of boundary condition and geometrical

parameter, the frequency parameter of the FGP elliptic cylindrical shell and panel decreases gradually with the increase of the porosity parameter.

**Table 5.** Frequencies $\Omega = \omega b \sqrt{\rho_1/E_1}$ of FGP elliptic cylindrical panel for different opening angle and porosity distribution.

| Type | $\theta_0$ | $e_0$ | | | | | | |
|---|---|---|---|---|---|---|---|---|
| | | 0.1 | 0.2 | 0.3 | 0.4 | 0.5 | 0.6 | 0.7 |
| Type 1 | $\pi/6$ | 4.998 | 4.940 | 4.882 | 4.825 | 4.768 | 4.717 | 4.675 |
| | $\pi/3$ | 1.287 | 1.273 | 1.259 | 1.246 | 1.233 | 1.223 | 1.216 |
| | $\pi/2$ | 0.267 | 0.265 | 0.264 | 0.263 | 0.263 | 0.263 | 0.264 |
| | $2\pi/3$ | 0.160 | 0.158 | 0.157 | 0.156 | 0.154 | 0.153 | 0.153 |
| | $5\pi/6$ | 0.146 | 0.145 | 0.144 | 0.143 | 0.143 | 0.142 | 0.143 |
| Type 2 | $\pi/6$ | 4.974 | 4.889 | 4.799 | 4.702 | 4.599 | 4.486 | 4.361 |
| | $\pi/3$ | 1.280 | 1.258 | 1.235 | 1.211 | 1.184 | 1.155 | 1.124 |
| | $\pi/2$ | 0.264 | 0.260 | 0.255 | 0.250 | 0.244 | 0.237 | 0.228 |
| | $2\pi/3$ | 0.159 | 0.156 | 0.153 | 0.150 | 0.147 | 0.143 | 0.139 |
| | $5\pi/6$ | 0.145 | 0.142 | 0.140 | 0.137 | 0.134 | 0.130 | 0.126 |
| Type 3 | $\pi/6$ | 4.931 | 4.800 | 4.660 | 4.509 | 4.346 | 4.165 | 3.962 |
| | $\pi/3$ | 1.269 | 1.235 | 1.199 | 1.160 | 1.118 | 1.072 | 1.019 |
| | $\pi/2$ | 0.261 | 0.254 | 0.247 | 0.239 | 0.230 | 0.221 | 0.210 |
| | $2\pi/3$ | 0.157 | 0.153 | 0.149 | 0.144 | 0.139 | 0.133 | 0.126 |
| | $5\pi/6$ | 0.143 | 0.139 | 0.135 | 0.131 | 0.126 | 0.121 | 0.115 |

Then, and finally, to help readers comprehend the vibration characteristics of such structures, the first three mode shapes of the FGP elliptic cylindrical shell and panel with various boundary conditions are given in Figure 3.

### 3.2. Dynamic Characteristics of FGP Elliptic Cylindrical Shell under Forced Vibration

Free vibration characteristics of the FGP elliptic cylindrical shell and panel are studied above. The forced responses of the FGP elliptic cylindrical shell and panel need to be studied next. Need to be mentioned that forced response includes steady-state response and transient response.

### 3.2.1. Steady-State Response in Frequency Domain

Before beginning the research of steady-state response in the frequency domain, it is necessary to carry out the correctness verification study of it. Figure 4 shows the comparison result between ABAQUS and the method in this paper. The detailed parameter settings of the finite element model are summarized as (1) The base feature used for the FE model is the shell shape; (2) the element shape is the Quad-dominated advancing front algorithm, and the global size is set as 0.0305 (dimensionless parameter). Furthermore, the material and geometrical parameters are the same as that used in the relative theoretical model.

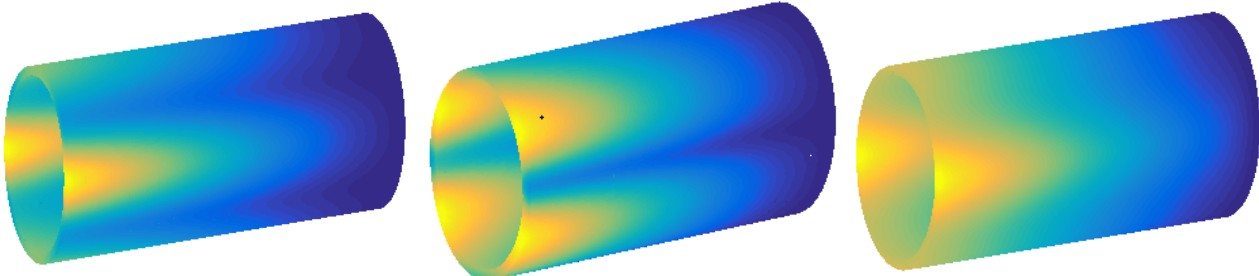

FGP elliptic cylindrical shell with CF.

**Figure 3.** *Cont.*

FGP elliptic cylindrical shell with SE2.

FGP elliptic cylindrical panel with SFSF.

FGP elliptic cylindrical panel with E3E3E3E3.

**Figure 3.** The first three mode shapes of FGP elliptic cylindrical shell and panel with various boundary conditions.

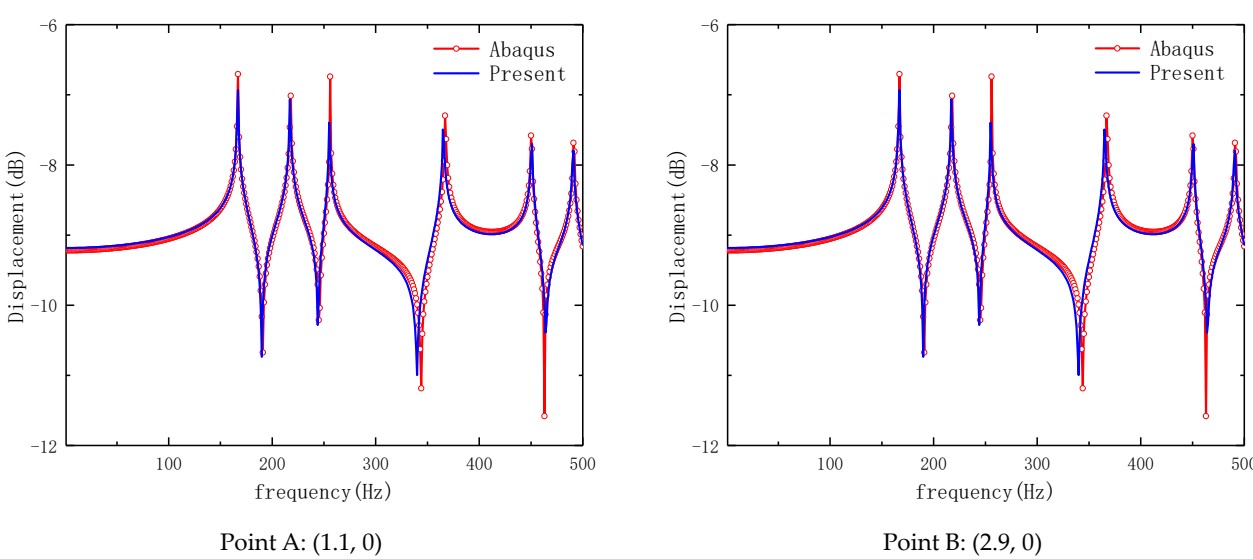

Point A: (1.1, 0)

Point B: (2.9, 0)

**Figure 4.** Comparison of the steady-state response of Type 1 FGP elliptic cylindrical shell with CC.

The loading form in Figure 4 is point load, and the loading amplitude value is $-1$ N, the specific form of the load is $f_w = \overline{f_w}\delta(\varphi - \varphi_0)\delta(\theta - \theta_0)$. We considered FGP elliptic cylindrical shell here. The structure parameters are as follows: $a = b = 1$ m, $L = 4$ m, $h = 0.1$ m, $e_0 = 0.2$. The loading location is $(x_0 = 2, \theta_0 = 0)$. Take two observation points on the structure, which are Point A: (1.1, 0) and Point B: (2.9, 0). The sweep frequency range is $f = 0$~500 Hz, $\Delta f = 1$ Hz. The boundary condition is set to the CC boundary condition. Through the comparison from this Figure, the method in this paper has a good agreement with the finite element calculation result on steady-state response. Therefore, the correction of this model has been verified.

The influence of boundary conditions on the steady-state response of the FGP elliptic cylindrical shell and panel is given in Figure 5. The geometrical parameters and loading parameters are as follows: The geometrical parameters of the FGP elliptic cylindrical shell are $a = 1$ m, $b = 2$ m, $L = 5$ m, and $e_0 = 0.2$. The geometrical parameters of the FGP elliptic cylindrical panel are $a = 1$ m, $b = 2$ m, $L = 5$ m, $\theta_0 = \pi$, and $e_0 = 0.2$. The load position is (1, 0), and the observation location is (3, 0), the sweep frequency range is $f = 0$ Hz~400 Hz, $\Delta f = 1$ Hz. Three types of boundary conditions are set to FGP elliptic cylindrical shell as follows, CF, E1E2, and E2E3, including both classical boundary conditions and elastic boundary conditions. And three types of boundary conditions are set to FGP elliptic cylindrical panel as follows, CFCF, E1E2E1E2, and E2E3E2E3, including both classical boundary conditions and elastic boundary conditions. It can be seen through this figure that the boundary condition of this structure has a direct relationship to the steady-state response of this structure, especially in low frequencies.

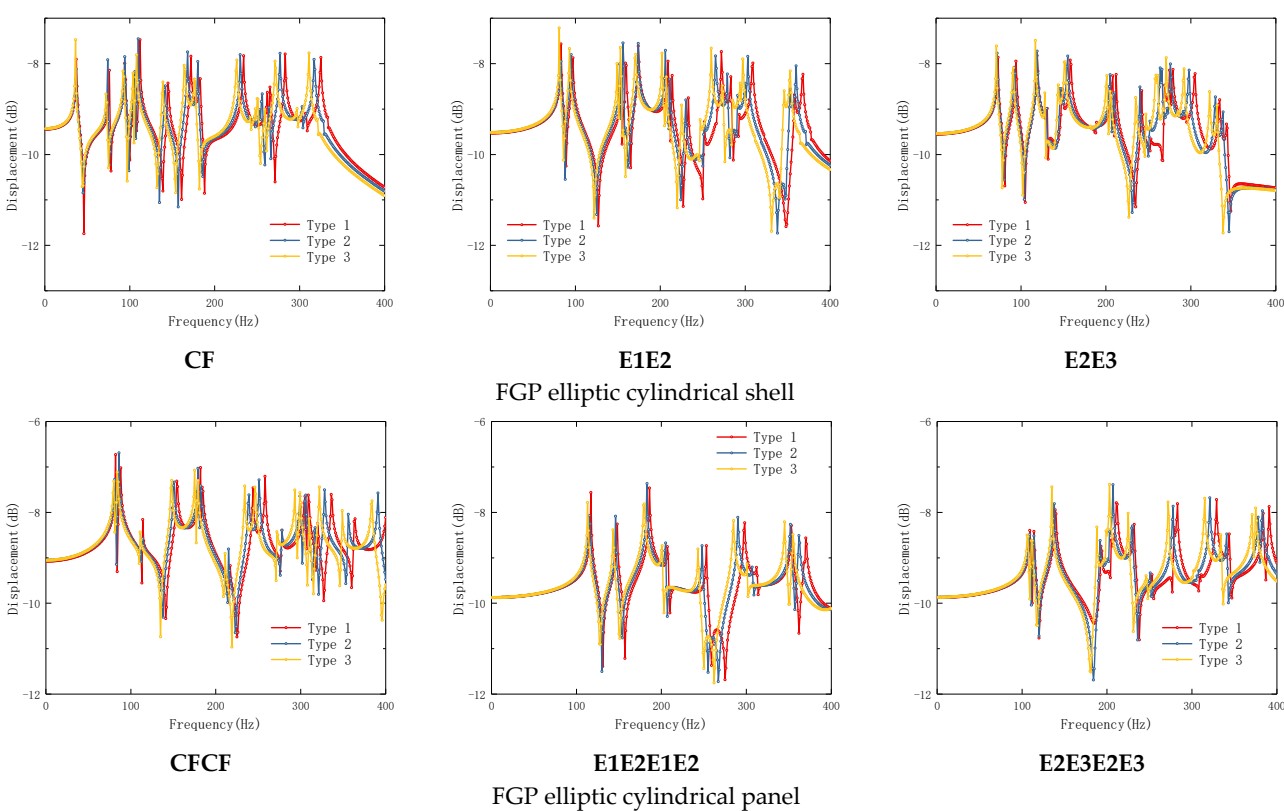

**Figure 5.** Effect of boundary conditions on the steady-state response of FGP elliptic cylindrical shell and panel with various boundary conditions.

For a further study on the influence of spring parameters on the steady-state response of this structure, specific research on the influence of spring parameters is given in Figure 6. Geometrical parameters and loading parameters are consistent with Figure 5. For boundary conditions, the boundary condition at $x = 0$ is set as a solid support boundary condition, and the boundary condition at $x = L$ is set as the elastic boundary condition. The boundary

conditions are divided into five groups as follows: Group 1: $k_u = 10^3, 10^8, 10^{14}$, $k_v = k_w = k_r = k_\theta = 10^{14}$; Group 2: $k_v = 10^3, 10^8, 10^{14}$, $k_u = k_w = k_r = k_\theta = 10^{14}$; Group 3: $k_w = 10^3, 10^8, 10^{14}$, $k_u = k_v = k_r = k_\theta = 10^{14}$; Group 4: $k_r = 10^3, 10^8, 10^{14}$, $k_u = k_v = k_w = k_\theta = 10^{14}$; and Group 5: $k_\theta = 10^3, 10^8, 10^{14}$, $k_u = k_v = k_w = k_r = 10^{14}$. It can be seen in Figure 6 that the influence of linear spring $k_w$ and rotation spring to steady-state response is very little, especially in low frequencies. However, linear spring $k_u$ and $k_v$ have a huge influence on the steady-state response of this structure. This is consistent with the results in the analysis of free vibration characteristics.

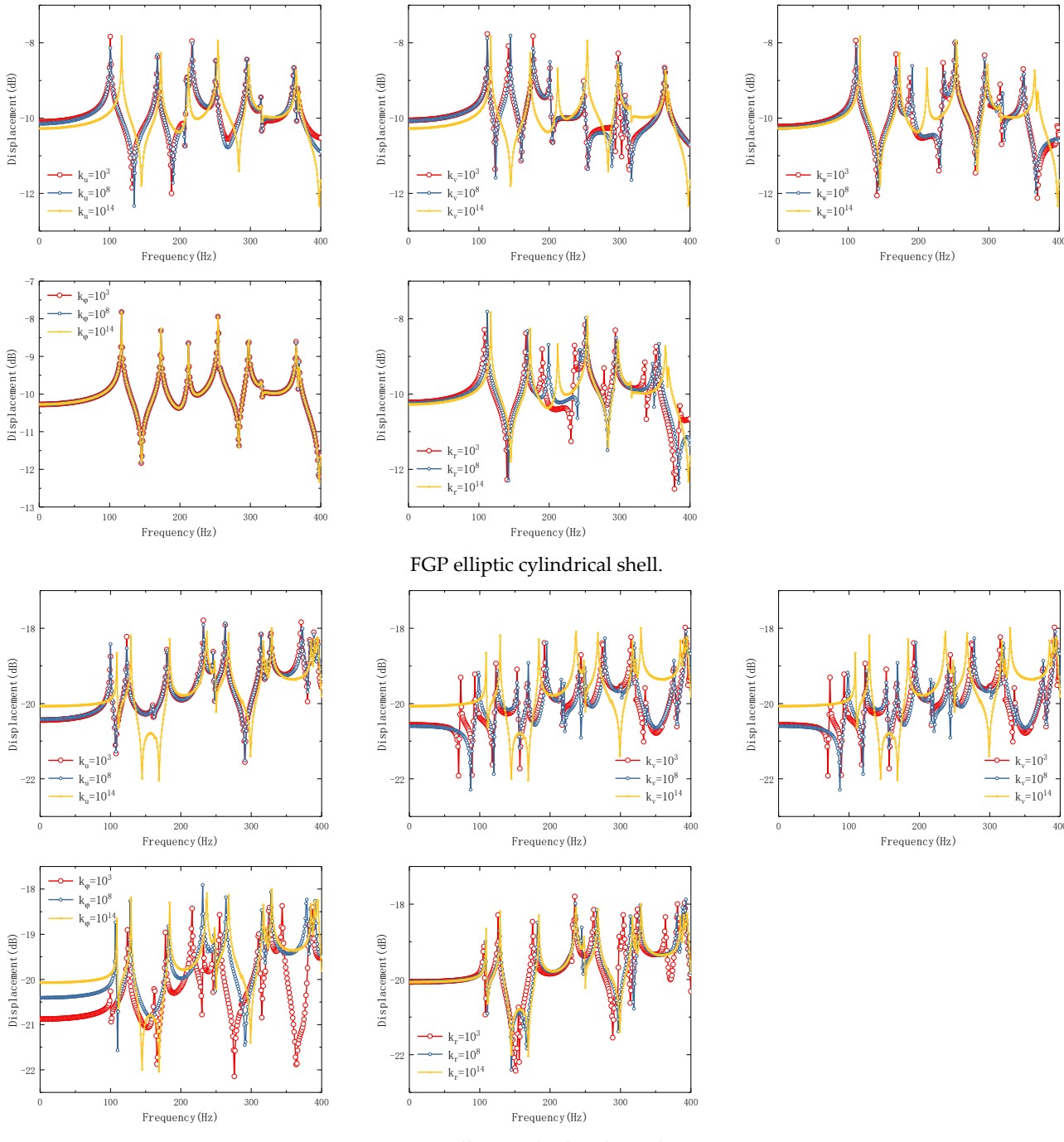

FGP elliptic cylindrical shell.

FGP elliptic cylindrical panel.

**Figure 6.** Effect of boundary spring parameters on the steady-state response of FGP elliptic cylindrical shell and panel.

The influence of thickness ratio on the steady-state response of FGP elliptic cylindrical shell and panel is studied in Figure 7. The geometrical parameters and loading parameters are consistent with Figure 6; only the thickness ratio changes. It can be seen from this figure that the change in thickness ratio not only influences the moving of the wave crest but also, as the increase of thickness ratio, the wave crest moves to the right gradually. In addition, with the increase in thickness ratio, the peak of the wave crest has also been affected. As the thickness ratio increases, the peak of the wave crest gradually decreases. The influence of porosity parameters on the steady-state response of the FGP elliptic cylindrical shell and panel is studied in Figure 8. The geometrical parameters and loading parameters are consistent with Figure 6. It can be seen from Figure 8 that no matter which kind of porosity distribution function, the wave crest moves to the left with the increase of porosity parameters.

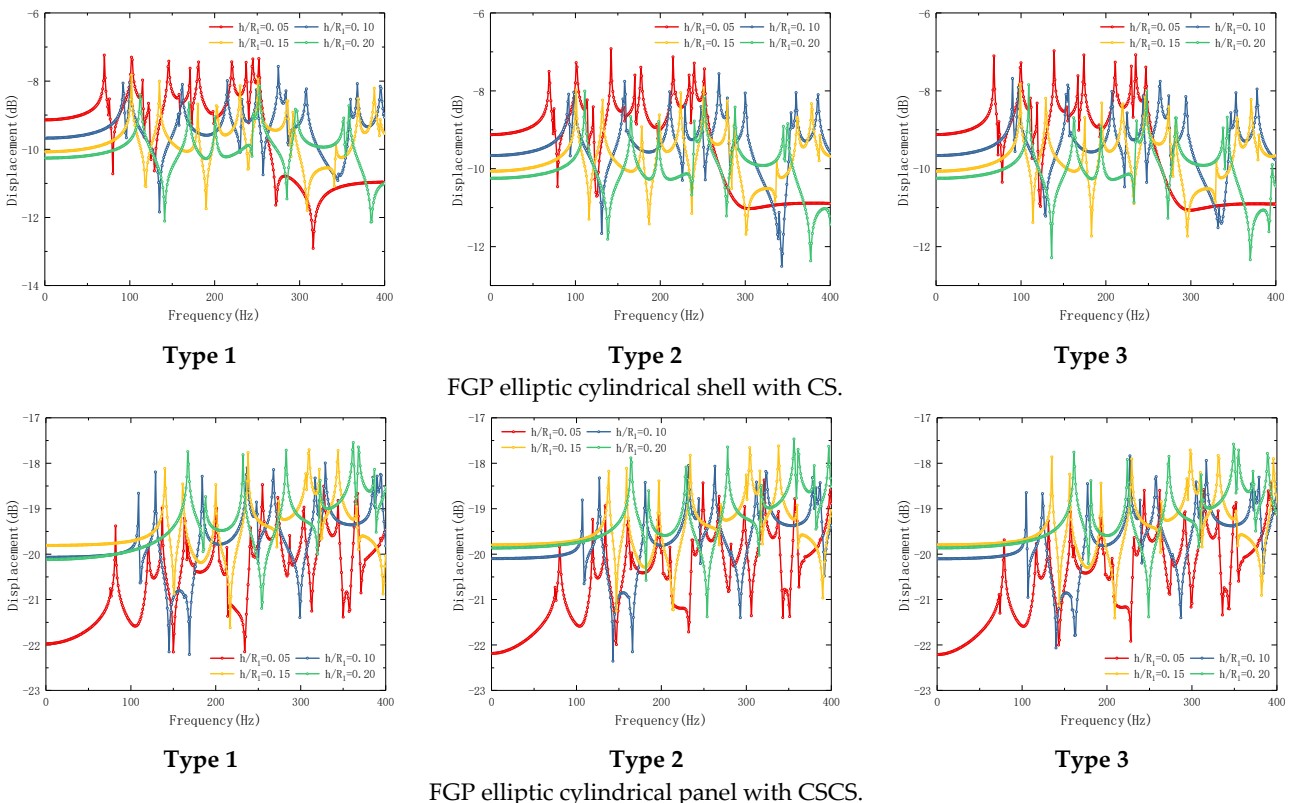

**Figure 7.** Effect of thickness ratio on the steady-state response of FGP elliptic cylindrical shell and panel.

The steady-state response in the frequency domain of the FGP elliptic cylindrical shell and panel has been studied above, and the transient response in the time domain will study below.

### 3.2.2. Transient Response in Time Domain

In this section, the systematic study of transient response is conducted. Before starting the studies of transient response, the correctness of the model in this paper on the transient response needs to be verified. The transient response of the FGP elliptic cylindrical shell at clamped boundary conditions between the method in this paper with finite element ABAQUS software has been given in Figure 9. The geometrical parameters are consistent with those in Figure 4. Here, the transient response load is the impact load, and the impact wave pulse is the rectangular pulse. The specific functions of impact load are given in Figure 10. The pulse functions in Figure 10 are as follows [22]:

$$\text{Rectangular pulse}: f(t) = \begin{cases} f_t & 0 \le t \le \tau \\ 0 & t > \tau \end{cases}$$

$$\text{Triangular pulse}: f(t) = \begin{cases} \frac{2t}{\tau}f_t & 0 \le t \le \frac{\tau}{2} \\ f_t - \frac{2}{\tau}\left(t - \frac{\tau}{2}\right)f_t & \frac{\tau}{2} \le t \le \tau \\ 0 & t > \tau \end{cases}$$

$$\text{Half} - \text{sine pulse}: f(t) = \begin{cases} f_t \sin\left(\frac{\pi t}{\tau}\right) & 0 \le t \le \tau \\ 0 & t > \tau \end{cases}$$

$$\text{Exponential pulse}: f(t) = \begin{cases} f_t e^{-\xi t} & 0 \le t \le \tau \\ 0 & t > \tau \end{cases}$$

where $f_t$ denotes the load amplitude, $\tau$ means the pulse width, and $t$ stands for the time variable.

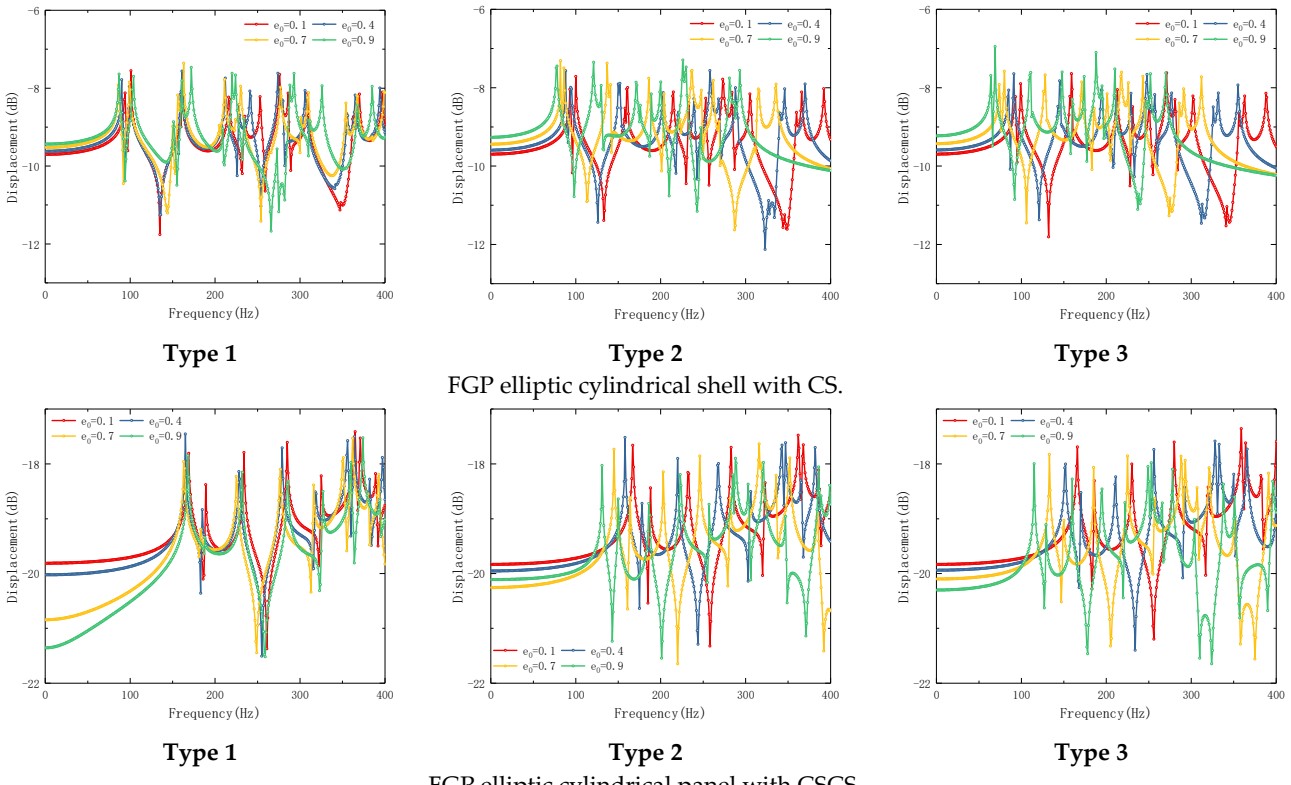

**Figure 8.** Effect of porosity ratios on the steady-state response of FGP elliptic cylindrical shell and panel.

In this calculation example, the load amplitude $f_t$ is $-1$ N, the total calculation time is 100 ms, and the impact time $\tau$ is also 100 ms, $\Delta t = 0.01$ ms. According to the comparison of Figure 9, it can be found that this method has excellent prediction accuracy for the transient response evaluation of the FGP elliptic cylindrical shell. Therefore, the parametric study of transient response will be carried out next.

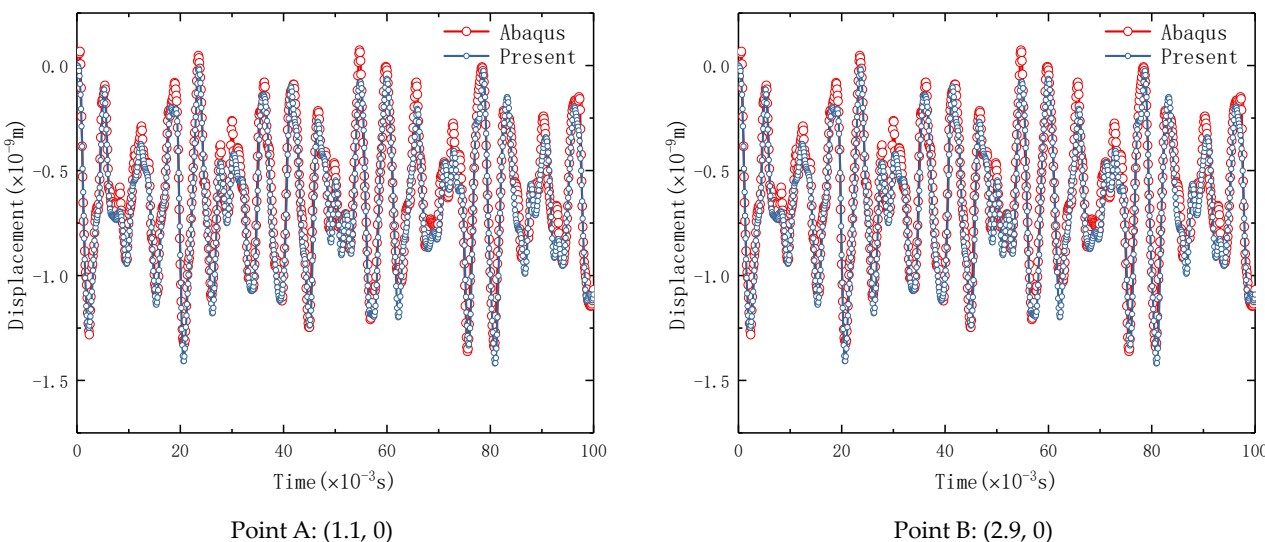

Point A: (1.1, 0)    Point B: (2.9, 0)

**Figure 9.** Comparison of the transient response of Type 1 FGP elliptic cylindrical shell with CC.

**Figure 10.** Sketch of load time domain curve.: (**a**) Rectangular pulse; (**b**) Triangular pulse; (**c**) Half-sine pulse; (**d**) Exponential pulse.

The influence of thickness ratio on the structure transient response of FGP elliptic cylindrical shell and panel is studied in Figure 11. The geometrical parameters in Figure 11 are consistent with those in Figure 7. The total calculation time and impact time $\tau$ are 100 ms. The load amplitude $f_t$ is $-1$ N. Load position is (1, 0), and the observation location is (4, 0). It can be seen from Figure 11 that the thickness ratio has a significant effect on the peak size of structure transient response, but for shock wave feedback, there is no real difference.

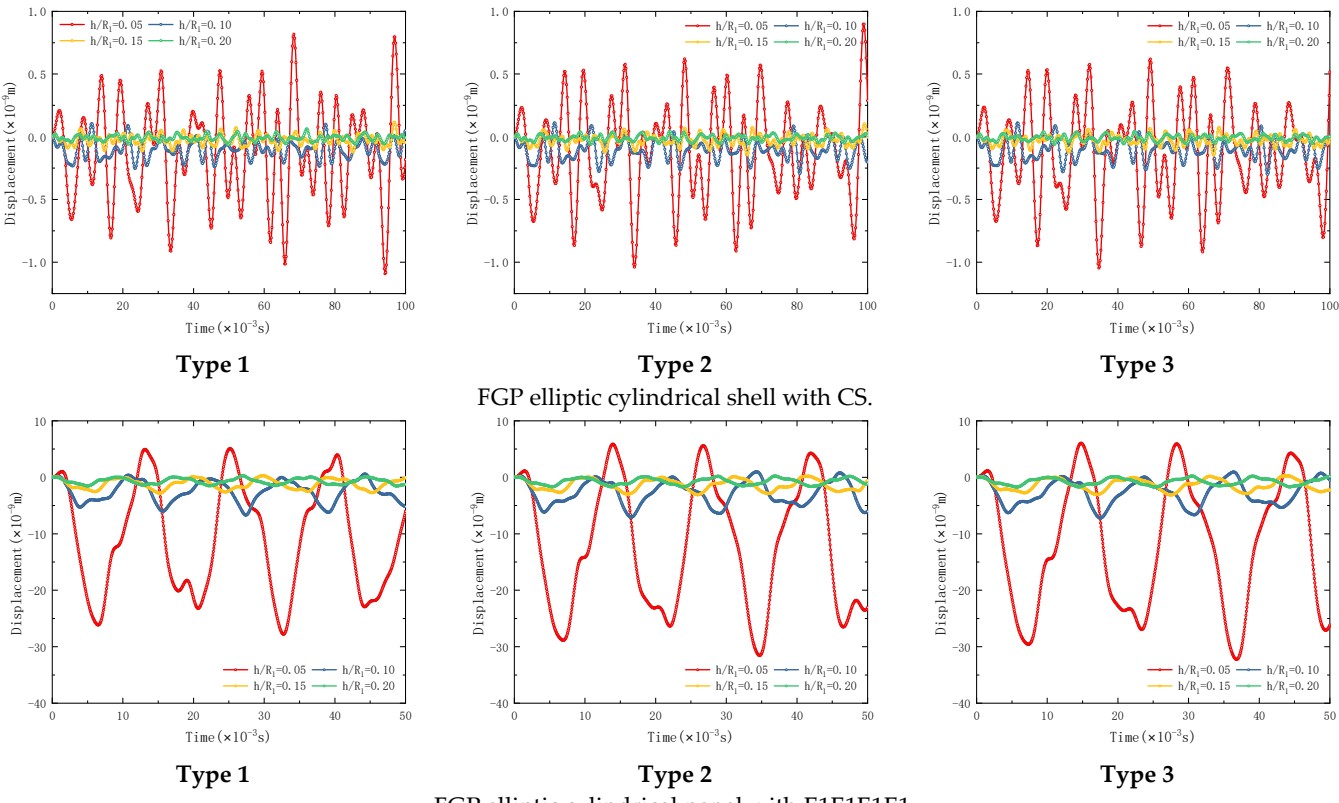

**Figure 11.** Effect of thickness ratio on the transient response of FGP elliptic cylindrical shell and panel.

The influence of the porosity parameter on the structure transient response of the FGP elliptic cylindrical shell and panel is studied in Figure 12. The geometrical parameters in Figure 12 are consistent with those in Figure 8. The total calculation time and impact time $\tau$ are 50 ms. The load amplitude $f_t$ is $-1$ N. Load position is (1, 0), and the observation location is (4, 0). It can be seen from Figure 12 that there is no significant delay in shock wave crest arrival time when the porosity parameter increases. It can only increase the peak size of the shock wave crest.

The influence of load types on the transient response of the FGP elliptic cylindrical shell and panel is studied in Figure 13. The geometrical parameters, impact load loading position, and observation location in Figure 13 are consistent with those in Figure 12. The calculation time settings are as follows: For FGP elliptic cylindrical shell, the total calculation time is 100 ms, and impact time $\tau$ is 40 ms, $\Delta t$ = 0.01 ms, and $f_t$ = $-1$ N; For FGP elliptic cylindrical panel, the total calculation time is 50 ms, and impact time $\tau$ is 20 ms, $\Delta t$ = 0.01 ms, and $f_t$ = $-1$ N. You can see from Figure 13, Triangular pulse and Half-sine pulse can significantly delay the arrival time of a shock wave and reduce the peak size of the shock wave crest.



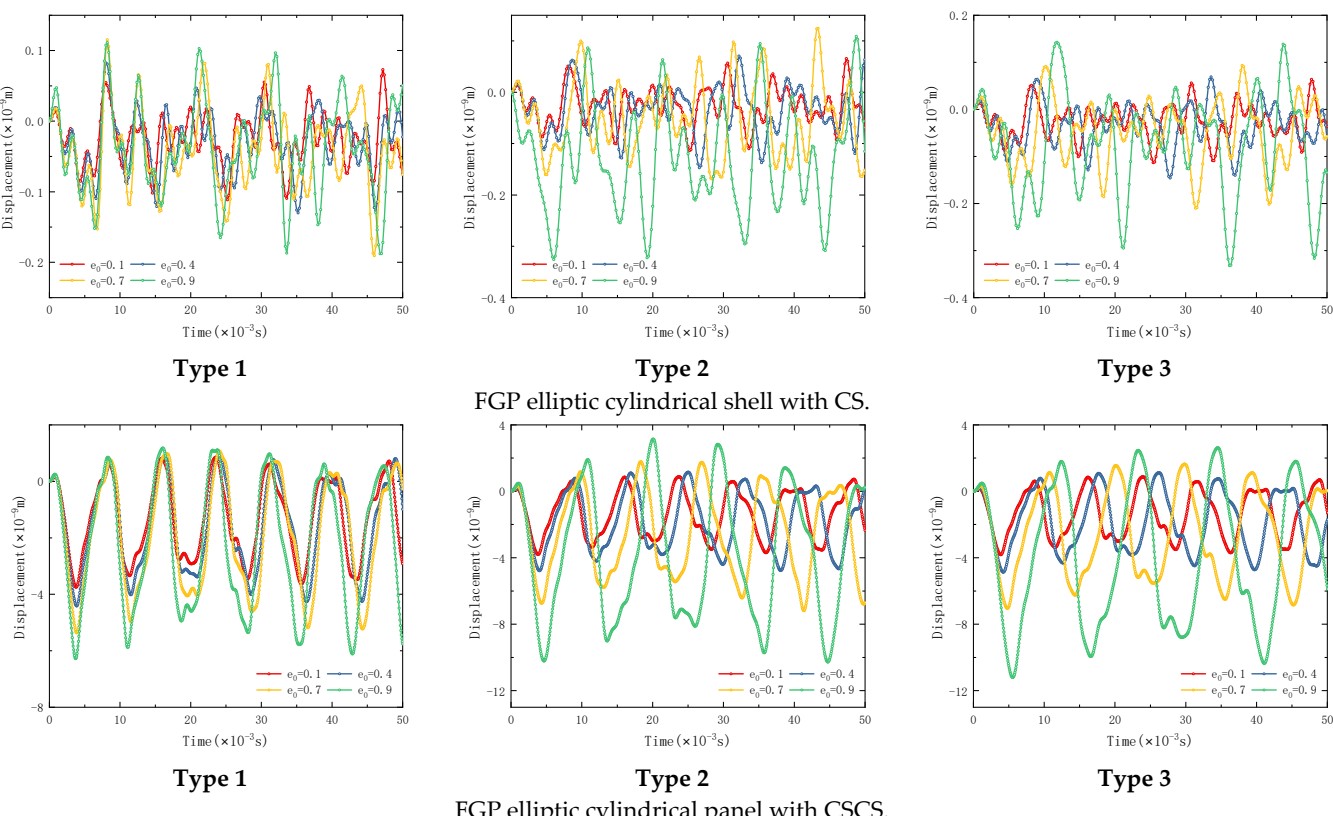

**Figure 12.** Effect of porosity ratios on the transient response of FGP elliptic cylindrical shell and panel.

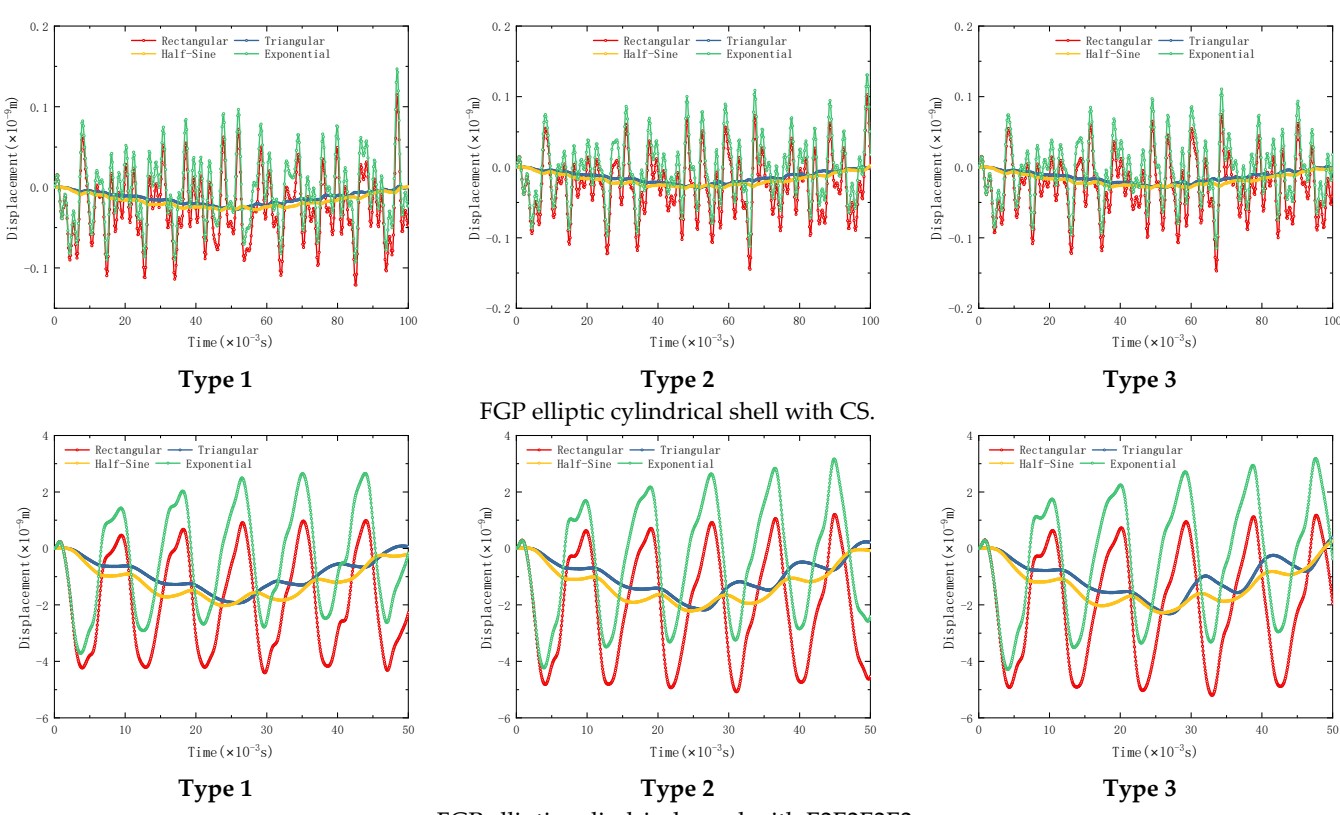

**Figure 13.** Effect of load types on the transient response of FGP elliptic cylindrical shell and panel.

## 4. Conclusions

In this paper, an FGP elliptic cylindrical shell and panel model was established, and three kinds of porosity distribution were considered. The energy expression of the FGP elliptic cylindrical shell and panel is established by FSTD theory. The artificial spring boundary technique was used for relaxing the boundary conditions, and the admissible displacement of the FGP structure was expressed as a combination of Jacobi orthogonal polynomials and Fourier series. The results from numerical analysis are as follows: For the free vibration characteristics of the FGP elliptic cylindrical shell and panel, the vibration characteristics of the structure can be regulated by changing the value of boundary stiffness. With the increase of radius ratio, length coefficient, and porosity coefficient, the structure frequency decreases. As the thickness coefficient increases, the structure frequency increases. As for the steady-state response in the frequency domain, linear spring $k_u$ and $k_v$ have a huge effect on the steady-state response. With the increase of thickness ratio, the wave crest moves to the right gradually, no matter which kind of porosity distribution function. With the increase of porosity, the wave crest moves to the left gradually. As for the transient response in the time domain, the thickness ratio has a significant effect on the peak size of the structure transient response, but for shock wave feedback, there is no real difference. There is no significant delay in shock wave crest arrival time when the porosity parameter increases, and it can only increase the peak size of the shock wave crest. Finally, the transient response of the FGP elliptic cylindrical shell and panel can be controlled via the unloading form of impact load.

**Author Contributions:** Conceptualization, J.L. and C.C.; methodology, J.L.; software, Q.Y.; validation, Q.Y., W.X. and K. Yang.; formal analysis, C.C.; investigation, C.C.; resources, J.L.; data curation, Z.M.; writing—original draft preparation, J.L.; writing—review and editing, Q.Y.; visualization, K.Y.; supervision, C.C.; project administration, W.X.; funding acquisition, W.X. All authors have read and agreed to the published version of the manuscript.

**Funding:** This Research was funded by Natural Science Foundation of Shandong Province (Grant No. ZR2022QE086); The Key Project of Shandong Province (Grant No. 2019GGX104035); The Project of Shandong Province Higher Educational Science and Technology Program (Grant No. J18KA035).

**Institutional Review Board Statement:** Not applicable.

**Informed Consent Statement:** Not applicable.

**Data Availability Statement:** Data available on request from the authors.

**Conflicts of Interest:** The authors declare no conflict of interest.

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
