# Peer review of "Dynamic Analysis of Functionally Graded Porous (FGP) Elliptic Cylindrical Shell Based on Jacobi Polynomials Ritz Method"

_jcs, doi:10.3390/jcs7080344_

Round 1
Reviewer 1 Report
A large part of the obtained results prove that the authors have worked hard. The main merit of this manuscript should be the idea of including and implementation of "Jacobi orthogonal polynomials" in approximation of the displacement field. However, the manuscript must seriously improved/completed or better rewritten to remove influences from reference articles. Considering this general aspect and also some particular aspects in the manuscript (see pdf file), this reviewer can not recommend the manuscript for publication in the present form. Below are some observations/suggestions.
General aspects:
-the manuscript is written hastily and carelessly with too many mistakes;
-the notations are not unified and many relations are incorrectly written;
-some quantities (parameters) are used without being defined in advance;
-some figures are incomplete or contain mistakes;
-some important implementation details related to the proposed model are not presented;
-some results, such as for example those of the dynamic response to various types of shock could be eliminated;
-personal contributions and advantages of the proposed method must be clearly highlighted.
Particular aspects:
-all noted by this reviewer are commented in the attached pdf file.
In conclusion, the paper appears to be deficient in the presentation and leaves some important aspects mostly unexplained. This reviewer cannot recommend its publication in this first form. The authors can consider the suggestion above as guidelines to improve the quality of the manuscript to make it useful to the readers.

Author Response
We are very glad to receive the reviewer's comments on our manuscript. The reviewers' comments are very helpful to improve our paper. We have revised the manuscript according to the reviewers' comments, and the revised parts have been highlighted in blue in the revised manuscript.

Reviewer 2 Report
Referee report on manuscript ``Dynamic analysis of functionally graded porous (FGP) elliptic cylindrical shell based on Jacobi polynomials-Ritz method’’ by J. Lu et al.
Authors analyze the dynamic characteristics of functionally graded porous by means of variational method and orthogonal polynomials presentation of the admissible displacements. The free vibration characteristic of FGP structure is analyzed, and forced response of FGP elliptic cylindrical shell and panel is studied. The accuracy of the model with a finite set of polynomials is verified by comparing with literature and ABAQUS software.
I have many remarks for this work.
1. Inaccuracies in formulas and in formulas comments.
Page 2 in bottom contains twice text "FGP elliptic cylindrical panels can be described by this geometric characteristic".
Page 3 eq. (5), (6) should be moved up after eq (4) before the explanations "where ..". rho_1, rho_2 should be instead of P_1, P_2.
Page 4 near center. "FSTD" should be changed on "first-order shear deformation shell theory (FSTD)"
Page 7 eq. (25) contains unknown bottom index j; explicit dependence of left side functions for i-segment from (x,theta,t) can be added. The corresponding Jacobi coeff. should contain both bottom index m,n.
Page 8. Is the coupling parameter 10^{14} true?
Page 23 Fig.1. h (thikness?) and coeff. k_v, k_w sould be explained. Also x,z coordinates on Fig.1 not correspond to coordinates x,z in the text.
2. In need more explanations for serious remarks.
Page 2 "artificial spring boundary technique", please give the reference and more explanations (for Fig.1).
Page 4 eqs. (7), (8) are not obvious. Should be explained or referenced.
Page 7 item 2.3. Please explain the selection of Jacoby polynomials instead of cos and sin along the theta angle.
Page 8. Jacobi polynomials with parameters alpha, beta=0 are Legendre polynomials.
3. I have style remarks.
Page 8 about the numerical method for the ordinary differential eqs. system. Text should contains terms "implicit method" and "unconditionally stable second order method" for selected parameters alpha and delta.
The corrected manuscript can be published in J. Compos. Sci.
Author Response
The response letter has uploaded as attached file

Reviewer 3 Report
1. Overview and general recommendation:
In this study, the authors investigated dynamic characteristics of functionally graded porous (FGP) elliptic cylindrical shells and panels. They developed models, considered different porosity distribution, explored the effect of various parameters, and verified the model through comparison.
Overall, the topic is interesting and it fits Journal of Composites Science. However, there are a few things that still need to be addressed. Therefore, I recommend that major revision is warranted.
2.2. Detailed comments:
General comments: The English and organization need native language editing. Although the contents are of great importance, the writing is relatively poor.
Also, please make sure to add line numbers. It’s very hard for reviewers to point out exact lines for comments.
Detailed comments are listed below.
(1) Page 1, Abstract. In addition to listing what you did, please also include some key findings in Abstract.
(2) Pages 1-2, Introduction. You cited plenty of papers, but just described what previous researchers did, which is not enough. You should briefly introduce their findings as well. The research gap remains unclear to readers, leaving the novelty of this work questionable.
(3) Page 2. “As mentioned above, the dynamic characteristics of the FGP structure are necessary to study.” à In fact, the importance of this work is not clear. You need to further explain this.
(4) Page 3, Line 11. Add “and” before “the minimum value”. Please carefully check throughout the paper and correct similar mistakes.
(5) Page 8. “For the numerical studies of the FGP elliptic cylindrical shell and panel below, it is important to mention that the calculation conditions are set as: The coupling parameter between FGP elliptic cylindrical shell and panel segments is set as 1014; The number of segments of FGP elliptic cylindrical shell and panel segments is set to 6; Jacobi expansion items are set as 8; Jacobi parameters α and β are both set to 0.” à Why did you set the conditions to these values? Any explanations?
(6) Page 9, The second paragraph “Table 1 shows …”. It seems you compared your results with references as you mentioned “in the reference” twice, but you did not cite any of them. Please do that.
(7) Page 10. “The first three mode shapes of the FGP elliptic cylindrical shell and panel with various boundary conditions are given in Fig. 3.” à Here you put “first THREE mode shapes”, but it says “First TWO mode shapes” in the caption of Fig. 3 (Page 25).
(8) Pages 11-12. Change title number “3.3.” and “3.4.” to “3.2.1.” and “3.2.2.” respectively.
Author Response

(The authors gave the same response as above.)

Round 2
Reviewer 1 Report
The results are difficult to understand due to the very large volume of analyzed cases.

Author Response
Letter of Response
Submission No.: jcs-2139833
Title: Dynamic analysis of functionally graded porous (FGP) elliptic cylindrical shell based on Jacobi polynomials-Ritz method
Date: 05/08/2023
Dear Editor(s) of Journal of Composites Science,
We are very glad to receive the reviewer's comments on our manuscript. The reviewers' comments are very helpful to improve our paper. We have revised the manuscript according to the reviewers' comments, and the revised parts have been highlighted in blue in the revised manuscript. Our responses to the reviewers' comments are also shown below:
Response to Reviewer #1:
Comment 1: Remove! only two coauthors.
Response 1: Thanks for the reviewer's comments. We have corrected this mistake.
Then, Wang and Wu [11] analyzed the dynamic behavior of an FGP cylindrical shell subject to different sets of immovable boundary conditions under free vi-bration.
Comment 2: et al.
Response 2: Thanks for the reviewer's comments. We have corrected this mistake.
Mirjavadi et al. [12] analyzed the buckling and nonlinear vibration of the FGP nanobeam.
Comment 3: a few words about units
Response 3: Thanks for the reviewer's comments. We have added the units in the revised manuscript as
The coupling springs of the FGP elliptic cylindrical shell and panel segments are set as 1014 N/m;
Comment 4: units!
Response 4: Thanks for the reviewer's comments. We have added the units in the revised manuscript as
The element shape is the Quad-dominated advancing front algorithm, and the global size is set as 0.0305 (dimensionless parameter).
Comment 4: Why blue?
Response 4: Thanks for the reviewer's comments. We have corrected this mistake.
[7] D. Chen, J. Yang, S. Kitipornchai, Elastic buckling and static bending of shear deformable functionally graded porous beam, Composite Structures, 133 (2015) 54-61.
Comment 5: Very small value, m/N ??
Response 5: Thanks for the reviewer's comments. The amplitude is related to the applied force. In this case, we adopted force -1N to stimulate the system. Therefore, the amplitude of the response is small. We have added the applied force on Page 13 of the revised manuscript as
In this calculation example, the load amplitude ft is -1N, the total calculation time is 100 ms, and the impact time τ is also 100 ms, Δt=0.01 ms.
The total calculation time and impact time τ are 100 ms. The load amplitude ft is -1N. Load position is (1, 0), and observation location is (4, 0).
The total calculation time and impact time ? are 50 ms. The load amplitude ft is -1N. Load position is (1, 0), and observation location is (4, 0).
For FGP elliptic cylindrical shell, total calculation time is 100 ms, and impact time ? is 40 ms, Δt=0.01 ms, and ft=-1N; For FGP elliptic cylindrical panel, total calculation time is 50 ms, and impact time ? is 20 ms, Δt=0.01 ms, and ft=-1N.
After an in-depth amendment to the manuscript, I would like to ask for your further review. We would be very grateful if you could kindly accept this paper for publication in of Journal of Composites Science.
Many thanks with regards
Cvinchiu (Corresponding author)
Professor
School of College of Mechanical & Electrical Engineering
Zaozhuang University
Zaozhuang 227101
- R. China
Email: cvinchiu@163.com

Reviewer 3 Report
Thanks for addressing the comments. I think the manuscript is good to be published in present form.
Author Response
Thanks for the reviewer's comments.
Round 3
Reviewer 1 Report
only one obs.: see attached file
